# Mast4 determines the cell fate of MSCs for bone and cartilage development

Pyunggang Kim[1,2,17], Jinah Park [1,3,17], Dong-Joon Lee [4,17], Seiya Mizuno[5,17], Masahiro Shinohara[6], Chang Pyo Hong[7], Yealeen Jeong[1], Rebecca Yun[1], Hyeyeon Park[1], Sujin Park[1], Kyung-Min Yang[8], Min-Jung Lee[4], Seung Pil Jang[9], Hyun-Yi Kim[4,10], Seung-Jun Lee [4], Sun U. Song[11,12], Kyung-Soon Park [2], Mikako Tanaka[13,14], Hayato Ohshima [13], Jin Won Cho[15], Fumihiro Sugiyama[5], Satoru Takahashi [5,18], Han-Sung Jung [4,18] & Seong-Jin Kim [1,8,16,18 ✉]

Mesenchymal stromal cells (MSCs) differentiation into different lineages is precisely controlled by signaling pathways. Given that protein kinases play a crucial role in signal transduction, here we show that Microtubule Associated Serine/Threonine Kinase Family Member 4 (Mast4) serves as an important mediator of TGF-β and Wnt signal transduction in regulating chondro-osteogenic differentiation of MSCs. Suppression of Mast4 by TGF-β1 led to increased Sox9 stability by blocking Mast4-induced Sox9 serine 494 phosphorylation and subsequent proteasomal degradation, ultimately enhancing chondrogenesis of MSCs. On the other hand, Mast4 protein, which stability was enhanced by Wnt-mediated inhibition of GSK-3β and subsequent Smurf1 recruitment, promoted β-catenin nuclear localization and Runx2 activity, increasing osteogenesis of MSCs. Consistently, $Mast4^{-/-}$ mice demonstrated excessive cartilage synthesis, while exhibiting osteoporotic phenotype. Interestingly, Mast4 depletion in MSCs facilitated cartilage formation and regeneration in vivo. Altogether, our findings uncover essential roles of Mast4 in determining the fate of MSC development into cartilage or bone.

[1] GILO Institute, GILO Foundation, Seoul 06668, Korea. [2] Department of Biomedical Science, College of Life Science, CHA University, Seongnam City463-400Kyunggi-do, Korea. [3] Amoris Bio Inc, Seoul 06668, Korea. [4] Division in Anatomy and Developmental Biology, Department of Oral Biology, Taste Research Center, Oral Science Research Center, BK21 FOUR Project, Yonsei University College of Dentistry, Seoul 03722, Korea. [5] Laboratory Animal Resource Center, University of Tsukuba, Tsukuba Ibaraki 305-8575, Japan. [6] Department of Rehabilitation for the Movement Functions, Research Institute, National Rehabilitation Center for Persons with Disabilities, Saitama 359-8555, Japan. [7] Theragen Bio Co., Ltd, Seongnam 13488, Korea. [8] Medpacto Inc., Seoul 06668, Korea. [9] Soo Hospital, Suwon 16249, Korea. [10] NGeneS Inc., Ansan-si 15495, Korea. [11] Research Institute, SCM Lifescience Inc., Incheon, Korea. [12] Department of Biomedical Sciences, Inha University College of Medicine, Incheon, Korea. [13] Division of Anatomy and Cell Biology of the Hard Tissue, Department of Tissue Regeneration and Reconstruction, Niigata University Graduate School of Medical and Dental Sciences, Niigata 951-8514, Japan. [14] Division of Dental Laboratory Technology, Meirin College, Niigata 950-2086, Japan. [15] Department of Systems Biology and Glycosylation Network Research Center, Yonsei University, Seoul, Korea. [16] TheragenEtex Co., Gyeonggi-do, Korea. [17]These authors contributed equally: Pyunggang Kim, Jinah Park, Dong-Joon Lee, Seiya Mizuno. [18]These authors jointly supervised this work: Satoru Takahashi, Han-Sung Jung, Seong-Jin Kim. ✉email: jasonsjkim@gilo.or.kr

Mesenchymal stromal cells (MSCs) are multipotent cells capable of differentiating into various lineages of mesenchymal cell types, including chondrocytes, osteoblasts, and adipocytes[1]. The commitment and differentiation of MSCs to each individual cell type depends on a variety of signaling pathways, including Wnt, TGF-β, BMP, and FGF[2]. During differentiation, the coordinated up-regulation and suppression of transcription factors are triggered via specific signaling pathways as well as interactions with other numerous transcription factors that act as co-regulators.

Sox9, a member of the family of high-mobility group (HMG) domain transcription factors, is an activator of chondrogenesis and regulates from the initiation of pre-cartilaginous condensations to the terminal differentiation of chondrocytes[3–5]. Sox9 activates collagen genes (*Col2, Col9, Col11*) and cartilage matrix genes (*Acan* and *Comp*) through the direct binding on their enhancers and promoters[6,7]. Considering Sox9 as a key regulator of chondrogenesis, Sox9 is strictly regulated by diverse mechanisms[8]. Several studies have reported phosphorylation events that regulate Sox9 in chondrocytes[9–11].

TGF-β signaling is involved in cartilage development and maintenance, especially stimulating chondrocyte differentiation at the early stage of chondrogenesis[12,13]. Animal studies have demonstrated that Smad3, a key mediator of TGF-β1 signaling, is required for maintaining articular cartilage, and mice with either Smad3-deficiency or chondrocyte-specific depletion of Smad3 resulted in degeneration of articular cartilage[14,15]. In addition, previous studies have reported that TGF-β1 signaling facilitates chondrogenesis through regulation of Sox9 in both Smad3-dependent and -independent manners[16–18], implying that TGF-β1-Sox9 axis is critical in regulating chondrogenesis.

Wnt/β-catenin signaling plays a crucial role in endochondral ossification by regulating osteoblast differentiation and maturation[19]. Wnt-induced stabilization of intracellular β-catenin and subsequent nuclear translocation leads to the activation of Runx2, a master transcription factor of osteoblast differentiation, especially in mesenchymal cells for development into bone[20]. Moreover, GSK-3β, a key negative regulator of canonical Wnt/β-catenin signaling, has shown to attenuate Runx2 activity during osteogenesis, suggesting GSK-3β as a potential molecular target for the treatment of bone diseases[21].

In this work, considering that protein kinases play a crucial role in signal transduction, we have sought to identify a gene that may be involved in the regulation of switching mesenchymal progenitor cells to specific lineages downstream of TGF-β or Wnt signals. Here, we identify that microtubule-associated serine/threonine kinase 4 (Mast4), which is suppressed by TGF-β1 during chondrogenesis of MSCs and enhanced by Wnt-mediated GSK-3β inhibition during osteogenesis of MSCs, plays an essential role in determining the cell fate of MSCs into chondrocyte or osteoblast differentiation. We show that Mast4-induced Sox9 phosphorylation at serine 494 residue results in proteasomal degradation of Sox9. We further demonstrate that Mast4 deficiency leads to increased Sox9 stability and Smad3-Sox9 association, which results in increased transcriptional activity of Sox9 and subsequent expression of chondrocyte marker genes, ultimately facilitating chondrogenic differentiation of MSCs. On the other hand, we find that GSK-3β-induced Mast4 phosphorylation triggers Mast4 recruitment of E3 ligase Smurf1, resulting in Mast4 degradation. We then show that Mast4 stabilized by Wnt-mediated GSK-3β inhibition promotes β-catenin nuclear localization, ultimately increasing Runx2 transcriptional activity and subsequent osteogenic differentiation of MSCs. The effects of Mast4 on chondro-osteogenesis of mesenchymal progenitors are confirmed in vivo by demonstrating excessive cartilage synthesis but osteoporotic or reduced bone formation in *Mast4*$^{-/-}$ mice. Interestingly, Mast4 depletion in MSCs facilitates cartilage formation and regeneration in vivo. Altogether, our findings uncover the essential roles of Mast4 in determining the fate of MSC development into cartilage or bone.

## Results

**Mast4 regulates chondro-osteogenic gene expression of MSCs.** We identified microtubule-associated serine/threonine kinase 4 (Mast4) as one of the genes down-regulated during chondrogenic differentiation of C3H10T1/2 murine mesenchymal stromal cells and ATDC5 murine chondrogenic cells (Fig. 1a and Supplementary Fig. 1). To investigate the role of Mast4 in MSC differentiation, we used the CRISPR/Cas9 system to disrupt *Mast4* gene in C3H10T1/2 cells that underwent chondrogenic differentiation with the addition of BMP-2 or TGF-β1 (Supplementary Fig. 2a, b)[22]. We observed increased expression of cartilage-specific genes and reduced expression of matrix metallopeptidase (Mmp)−9/13 by CRISPR/Cas9-mediated Mast4 depletion and shRNA-mediated Mast4 knockdown in undifferentiated C3H10T1/2 cells (Supplementary Fig. 2c, d). Furthermore, during differentiation to chondrocytes under BMP-2 stimulation in high-density micromass cultures, induction of cartilage matrix genes occurred earlier and was enhanced in Mast4-depleted C3H10T1/2 cells (Fig. 1b and Supplementary Fig. 2d). Interestingly, we noticed that Mast4 depletion-mediated increase of Sox9 protein expression was more evident than that of Sox9 mRNA expression, implying post-translational regulation of Sox9 by Mast4 (Supplementary Fig. 3). Since Sox9 is a key transcription factor for chondrocyte-specific genes, an increased amount of Sox9 protein induced by Mast4 depletion may help maintain the chondrocytic character of C3H10T1/2 cells. Next, we generated Mast4-overexpressing C3H10T1/2 cells to examine whether Mast4 inhibits chondrogenesis. Due to high molecular weight (>285 kDa) and relatively low expression of full-length Mast4, we alternatively used a truncated Mast4 construct (Mast4-PDZ) that contained a DUF, kinase, and PDZ domain (Supplementary Fig. 4a). We found that forced expression of Mast4-PDZ protein efficiently suppressed the expression of chondrocyte marker genes (Supplementary Fig. 4b).

We further characterized the enhancement of chondrogenesis induced by Mast4 depletion by performing RNA sequencing using wild-type and Mast4-depleted C3H10T1/2 micromass cultures treated with BMP-2 for 6 days. Differentially expressed gene (DEG) analysis identified 151 up-regulated genes and 220 down-regulated genes in Mast4-depleted C3H10T1/2 cells (Supplementary Fig. 5). Gene ontology enrichment analysis and the heatmap showed that genes related to chondrocyte differentiation (FDR = $1.01 \times 10^{-3}$) and cartilage development (FDR = $5.05 \times 10^{-6}$) were significantly enriched in Mast4-depleted cells (Supplementary Fig. 5 and Fig. 1c). The expression of upregulated genes involved in cartilage development (Fig. 1d) and down-regulated genes involved in bone formation (Fig. 1e) in the chondrogenic and osteogenic differentiated Mast4-depleted cells, respectively, were confirmed by qRT-PCR. Furthermore, among collagen genes, cartilage-specific collagen genes (*Col2a1, Col9a1, Col9a2, Col9a3, Col11a1* and *Col11a2*) were specifically up-regulated by Mast4 depletion (Supplementary Fig. 6). Particularly, 21 significantly up-regulated genes, related to cartilage and chondrocyte development, in Mast4-depleted cells highly interacted with the genes associated with BMP and TGF-β signaling pathways, which play important roles in chondrogenesis, besides cartilage development in a transcriptional network (Fig. 1f). Interestingly, *Sox9* directly interacted with *BMP2* and *TGF-β1*, being appeared to function as the hub of the network as larger nodes in the two signaling pathways. In summary, our data

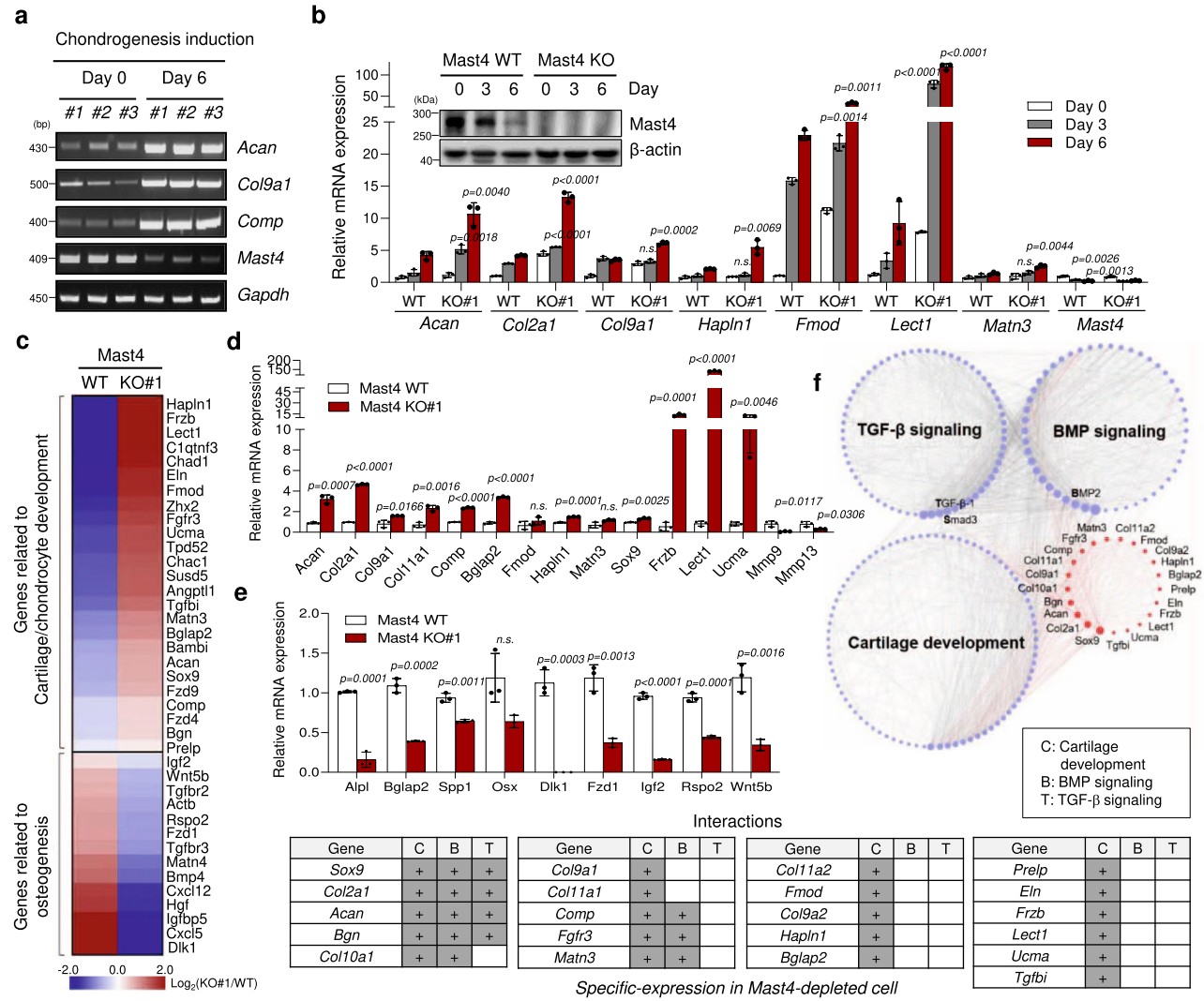

**Fig. 1 Targeted deletion of the *Mast4* gene enhances cartilage matrix gene expression and reduces osteogenic gene expression in vitro.**
**a** Representative RT-PCR result, obtained from at least three independent experiments, of high-density micromass culture of C3H10T1/2 cells in the presence of BMP-2. **b** Representative qRT-PCR result of the expression of chondrocyte marker genes in the differentiating wild-type and Mast4-depleted C3H10T1/2 cells. Mast4 protein expression was confirmed by western blot during chondrogenic differentiation. **c** Heatmap of DEGs, classified under cartilage/chondrocyte development and osteogenesis, of chondrogenic differentiated wild-type and Mast4-depleted C3H10T1/2 cells for 6 days. **d** Representative qRT-PCR result of Sox9-targeted genes and *Mmp9/13*, which were identified by RNA sequencing, in wild-type and Mast4-depleted C3H10T1/2 cells differentiated to chondrocytes for 6 days. **e** Representative qRT-PCR result of osteoblast marker genes and the genes related to osteogenesis, which were identified in **c**, in wild-type and Mast4-depleted C3H10T1/2 cells differentiated to osteoblasts for 10 days. **f** Transcriptional network of the DEGs related to cartilage and chondrocyte development, TGF-β signaling, and BMP signaling in the wild-type and Mast4-depleted C3H10T1/2 cells differentiated to chondrocytes for 6 days. The node size was set on the basis of connectivity of the nodes. Red-colored circles indicate genes showing differential expression in Mast4-depleted C3H10T1/2 cells. **b**, **d** and **e** Data are representative mean ± SD of three independent experiments, each conducted in triplicate ($n = 3$). Unpaired two-tailed Student's t test ($P < 0.05$) with Benjamini-Hochberg correction for multiple tests was conducted for all statistical analyses. *P* values versus WT at corresponding day; n.s. not significant. Uncropped blots in **a**, **b** are shown in Source Data file. Source data for **b**, **d** and **e** are provided as a Source Data file.

suggest that Mast4 may play a key role in regulating the chondrogenic differentiation of MSCs.

**Mast4 modulates chondrogenesis by regulating Sox9 stability.**
Next, we validated the effect of Mast4 on chondrogenesis of MSCs (Supplementary Fig. 7a). The difference in cell growth was not significant between the wild-type and Mast4-depleted cells, whereas the cell growth was significantly increased in Mast4-PDZ-overexpressing C3H10T1/2 cells (Supplementary Fig. 7b). However, alcian blue staining demonstrated that chondrogenic differentiation in Mast4-depleted cells was enhanced, whereas overexpression of Mast4-PDZ suppressed chondrogenic

differentiation (Fig. 2a and Supplementary Fig. 7c). The 3D spheroid formation assay using low-binding plates[23] also showed increased spheroid size and expression of cartilage-specific genes by Mast4 depletion (Supplementary Fig. 7d, e). In addition, chondrogenic differentiation of MAST4-depleted human bone marrow-derived stem cells (hBMSC) produced significantly increased size of cartilaginous aggregates and increased mRNA expression of *COL2A1* and *ACAN* (Fig. 2b and Supplementary Fig. 8). Chromatin immunoprecipitation (ChIP) assays further revealed that Mast4 deficiency significantly increased the binding of Sox9 protein to the promoter region of *Col2a1* gene, which was reduced by Mast4 overexpression, during chondrogenic

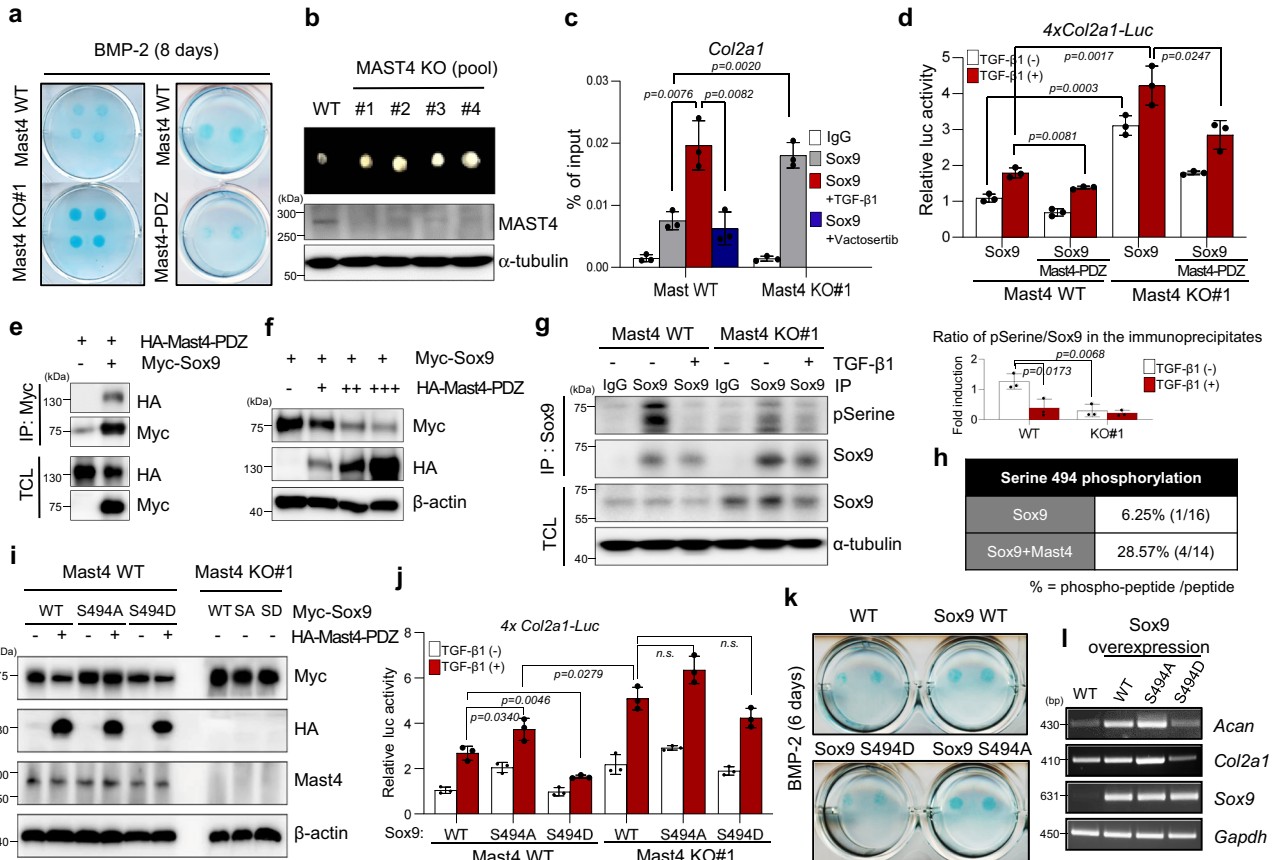

**Fig. 2 Mast4 modulates chondrogenesis through post-translational regulation of Sox9. a** Alcian blue staining results of C3H10T1/2 cells. **b** $2×10^5$ of hBMSC were differentiated into chondrocytes for 21 days, followed by protein extraction from the pellets. **c** C3H10T1/2 cells were differentiated into chondrocytes for 6 days, followed by Sox9 ChIP on *Col2a1* gene (TGF-β1 (5 ng/ml) and Vactosertib (0.5 μM), a TGF-β type I receptor kinase inhibitor, for 48 h). **d** 4xCol2a1-luc, Sox9, and Mast4-PDZ were transiently overexpressed in the wild-type and Mast4-depleted C3H10T1/2 cells, followed by TGF-β1 treatment (3 ng/ml for 24 h). **e** Mast4-PDZ and Sox9 were transfected to C3H10T1/2 cells, followed by immunoprecipitation assay. **f** Western blot analysis of Sox9 stability in C3H10T1/2 cells. **g** Wild-type and Mast4-depleted C3H10T1/2 cells were differentiated to chondrocytes for 6 days, followed by TGF-β1 (5 ng/ml for 48 h) and MG-132 (10 μM for 6 h) treatment, and Sox9 immunoprecipitation. Band intensities representing pSerine and Sox9 were converted by densitometry using ImageJ. Data are reported as mean ± SD of three independent experiments ($n = 3$). Unpaired two-tailed Student's t test ($P < 0.05$) was conducted for statistical analysis. **h** Sox9 and Mast4-PDZ were transfected to C3H10T1/2 cells in the presence of MG-132 (10 μM for 6 h). Two independent MASS SPEC analyses were conducted. **i** Mast4-PDZ was co-transfected with Sox9 wild-type (WT), S494A, or S494D mutants into C3H10T1/2 cells. **j** 4xCol2a1-luc and Sox9 WT/S494A/S494D were co-transfected to C3H10T1/2 cells, followed by TGF-β1 treatment (3 ng/ml for 24 h). **k** Representative alcian blue staining of C3H10T1/2 cells stably overexpressing Sox9 WT, S494A, and S494D in micromass cultures for 6 days. **l** The wild-type and stably overexpressing Sox9 WT, S494A, and S494D C3H10T1/2 cells were differentiated into chondrocytes for 6 days. The mRNA expression was examined by RT-PCR. **c**, **d** and **j** Data are representative mean ± SD of three independent experiments, each conducted in triplicate ($n = 3$). Unpaired two-tailed Student's t test ($P < 0.05$) with Benjamini-Hochberg correction for multiple tests was conducted for all statistical analyses. n.s. not significant. **a**, **b**, **e**-**g**, **i**, and **l** The representative results were obtained from at least three independent experiments. TCL total cell lysates. Uncropped blots in **b**, **e**, **f**, **g** and **i** are shown in Source Data file. Source data for **c**, **d**, **g** and **j** are provided as a Source Data file.

differentiation (Supplementary Fig. 9a). Notably, Mast4 depletion was sufficient to considerably increase the binding of Sox9 protein to the promoter region of *Col2a1* gene even in the absence of TGF-β1 treatment, which activates *Col2a1* gene transcription through Sox9[18] (Fig. 2c). On the other hand, Mast4-PDZ overexpression reduced basal and TGF-β1-induced *Col2a1* promoter activity as well as Sox9-induced increase of *Col2a1* promoter activity, which was more noticeable in Mast4-deficient cells (Supplementary Fig. 9b and Fig. 2d).

Our observation of the increase of Sox9 protein expression by Mast4 depletion (Supplementary Fig. 3) prompted us to examine post-translational regulation of Sox9 protein stability by Mast4. Given that Mast4 functions as a serine/threonine kinase, we examined whether Mast4 induced Sox9 protein phosphorylation, leading to its protein degradation. We found that Mast4 bound to Sox9 and that overexpression of not only full-length but also

truncated Mast4 decreased Sox9 protein expression, which was restored by blockade of Mast4 expression by siRNA, in a dose-dependent manner (Figs. 2e, f and Supplementary Fig. 9c). As expected, Sox9 serine phosphorylation was significantly reduced in Mast4-deficient cells, while basal Sox9 protein expression was increased, suggesting that Mast4 might promote Sox9 degradation by inducing Sox9 phosphorylation (Fig. 2g and Supplementary Fig. 9d). Indeed, Mast4-induced Sox9 degradation was restored by MG-132 treatment (Supplementary Fig. 9e). To identify the Mast4-mediated phosphorylation sites in Sox9 protein, we transiently overexpressed Sox9 and Mast4-PDZ proteins in C3H10T1/2 cells in the presence of MG-132 to prevent Mast4-mediated Sox9 degradation. And then Sox9 was immunoprecipitated, followed by MASS SPEC analysis. MASS SPEC analysis revealed serine 494 whose phosphorylation status was regulated by Mast4 (Fig. 2h). Particularly, we observed that

Mast4-induced Sox9 degradation was more evident in C3H10T1/2 cells transfected with Sox9 WT and S494D substitution mutant of Sox9 protein which mimics phosphorylation, while S494A substitution mutant of Sox9 protein was barely degraded by Mast4 (Fig. 2i). We further found that luciferase activity of Sox9-induced *Col2a1* promoter reporter by Sox9 S494A was increased in the wild-type cells, comparable to that by Sox9 WT in Mast4-depleted cells (Fig. 2j). Interestingly, the difference in *Col2a1* promoter activities by Sox9 WT, S494A and S494D was not significant in the absence of Mast4. Besides, Mast4-PDZ-mediated reduction of *Col2a1* activity was not significant in Sox9 S494A-transfected cells (Supplementary Fig. 10). We further observed increased chondrogenesis of C3H10T1/2 cells stably overexpressing Sox9 S494A and decreased chondrogenesis of Sox9 S494D-overexpressing cells by alcian blue stain staining and RT-PCR for chondrocyte marker genes (Fig. 2k, l). These results indicate that Mast4-mediated Sox9 protein phosphorylation at serine 494 may play an important role in chondrogenic differentiation of MSCs.

**TGF-β-induced *Mast4* suppression enhances Sox9-Smad3 association**. We have observed the interaction between Mast4 protein and Smad3 protein as well as increased TGF-β1/Smad3-induced transcriptional activity in Mast4-deficient cells (Supplementary Fig. 11). In addition, a previous report demonstrated that Smad3 enhances Sox9-dependent transcriptional activation during chondrogenesis through interaction with Sox9[18]. Therefore, we hypothesized that Mast4 might suppress TGF-β1/Smad3-induced Sox9 transcriptional activity by facilitating Sox9 protein degradation, leading to decreased Smad3-Sox9 complex formation. Indeed, both Sox9 protein stability and Smad3-Sox9 association were increased by Mast4 depletion (Fig. 3a). Moreover, we found that Mast4 depletion elevated Smad3 occupancy at the Sox9-binding site of the *Col2a1* gene (Fig. 3b). Importantly, the binding of Smad3 to the Smad7 promoter and expression levels of TGF-β1 cytostatic target genes, such as *Cdkn1a*, *c-myc*, and *Smad7*, were not affected by the status of Mast4 expression, indicating that TGF-β1-induced chondrogenic differentiation may rely upon the level of Mast4 which controls Sox9 protein stability (Supplementary Fig. 12).

Considering the role of TGF-β1 signaling in chondrogenic differentiation[12], we investigated whether TGF-β1 induced chondrogenesis through regulation of Mast4 expression. Interestingly, TGF-β1 treatment markedly suppressed both mRNA and protein expression of Mast4 (Fig. 3c). Mast4 promoter activity was also suppressed by the TGF-β1 treatment, but Smad3 occupancy at the *Mast4* promoter was significantly increased by TGF-β1 treatment, implying that TGF-β1/Smad3 signaling may negatively regulate *Mast4* transcription (Fig. 3d, e and Supplementary Fig. 13). In particular, gradual reduction of Mast4 protein and mRNA expression level were observed during chondrogenic differentiation of C3H10T1/2 cells and in human primary chondrocytes treated with TGF-β1, respectively (Fig. 3f, g). These were well correlated with the increased Sox9 protein expression and mRNA expression of chondrocyte marker genes, reinforcing our findings of enhanced chondrogenesis of Mast4-deficient cells. Furthermore, inhibition of TGF-β signaling by the treatment of C3H10T1/2 cells with Vactosertib, a TGF-β receptor kinase inhibitor, prevented down-regulation of *Mast4* gene and blocked induction of chondrocyte marker genes, indicating that suppression of Mast4 expression by TGF-β1 is essential for chondrogenic differentiation of MSCs (Supplementary Fig. 14). Taken together, our observations suggest that TGF-β1/Smad3 signaling promotes chondrogenesis through suppression of Mast4 gene expression, resulting in subsequent accumulation of Sox9 protein.

**Wnt-induced Mast4 stabilization increases osteogenesis of MSC**. It is widely appreciated that Wnt/β-catenin signaling plays a critical role in skeletal development by governing the lineage commitment and differentiation of mesenchymal stromal cells into osteoblasts[19]. Our observation of down-regulation of the genes related to osteogenesis by Mast4 depletion in C3H10T1/2 cells led us to investigate whether Mast4 mediates Wnt/β-catenin-induced osteogenesis. Indeed, Alizarin Red S staining demonstrated enhanced osteogenic differentiation of C3H10T1/2 cells by stable overexpression of Mast4-PDZ (Fig. 4a and Supplementary Fig. 15a, b). Interestingly, Mast4 expression was elevated by Wnt3a stimulation in undifferentiated MC3T3-E1 pre-osteoblasts (Fig. 4b). Moreover, osteogenic differentiation of MC3T3-E1 cells led to increased Mast4 expression as well as increased Wnt signaling, shown by increased levels of active β-catenin and inactive GSK-3β (Supplementary Fig. 15c). We further observed that stable overexpression of Mast4-PDZ significantly increased β-catenin expression and subsequent Runx2 expression during osteogenic differentiation of C3H10T1/2 cells (Fig. 4c). These observations were supported by RNA-seq analysis, demonstrating significant down-regulation of genes associated with osteogenesis in Mast4-depleted C3H10T1/2 cells (Fig. 1c).

Having evidence for a role of Mast4 as a potent mediator of Wnt/β-catenin-induced osteogenic differentiation of progenitor cells, we found that Mast4 protein expression was decreased by GSK-3β in a dose-dependent manner and that GSK-3β inhibitor treatment dramatically increased Mast4 expression (Fig. 4d). Moreover, Mast4 degradation was significantly delayed by GSK-3β inhibition as well as by GSK-3β depletion (Supplementary Fig. 15d, e). Since GSK-3β-induced phosphorylation has been found to regulate the stability of numerous target proteins[24], we next addressed whether GSK-3β interacted with Mast4 to induce its degradation by phosphorylating Mast4. We observed that GSK-3β bound to the kinase domain of Mast4 and induced Mast4 serine phosphorylation (Supplementary Fig. 16a, b and Fig. 4e). Besides, among the reported E3 ubiquitin ligases that are recruited to target proteins in GSK3 phosphorylation-dependent manner[24], Smurf1 showed interaction with the kinase domain of Mast4 which intensity was regulated by GSK-3β status (Supplementary Fig. 16c and Fig. 4f). We further observed that Smurf1-mediated Mast4 degradation was enhanced by GSK-3β overexpression, but blocked by GSK-3β depletion (Fig. 4g). Considering that WW domain-containing E3 ligases, including Smurf1, recognize the consensus PY motif (PPxY)[25], we found that mutation of PY-like motif in the kinase domain of Mast4 (P628A/Y634A) prevented Smurf1-mediated Mast4 degradation, which was no longer affected by GSK-3β depletion (Fig. 4h). To map the region responsible for the GSK-3β-mediated degradation of Mast4 in conjunction with Smurf1, we have generated serial deletion mutants of Mast4-PDZ (a.a.1-a.a.1229). As shown in Fig. 4i, expression levels of Mast4 deletion mutants containing a region between a.a.620 and a.a.1229 of Mast4-PDZ were significantly lower than those of Mast4 deletion mutants shorter than a.a.620 in the presence of GSK-3β, suggesting that a region between a.a.620 and a.a.1229 might contain the expected GSK3 consensus phosphorylation site (SxxxSP). Interestingly, amino acid deletion at residues S632-S636 decreased Smurf1-mediated Mast4 polyubiquitination, which was enhanced by GSK-3β, while increasing Mast4 stability (Fig. 4j). Eventually, β-catenin nuclear localization and Runx2 transcriptional activity were increased by Mast4 overexpression and further enhanced by the blockade of GSK3β regulation of Mast4 through S632-S636 deletion (Fig. 4k, l). Furthermore, either Mast4-PDZ overexpression or GSK-3β depletion resulted in significantly increased Runx2 transcriptional activity and subsequent osteogenic differentiation

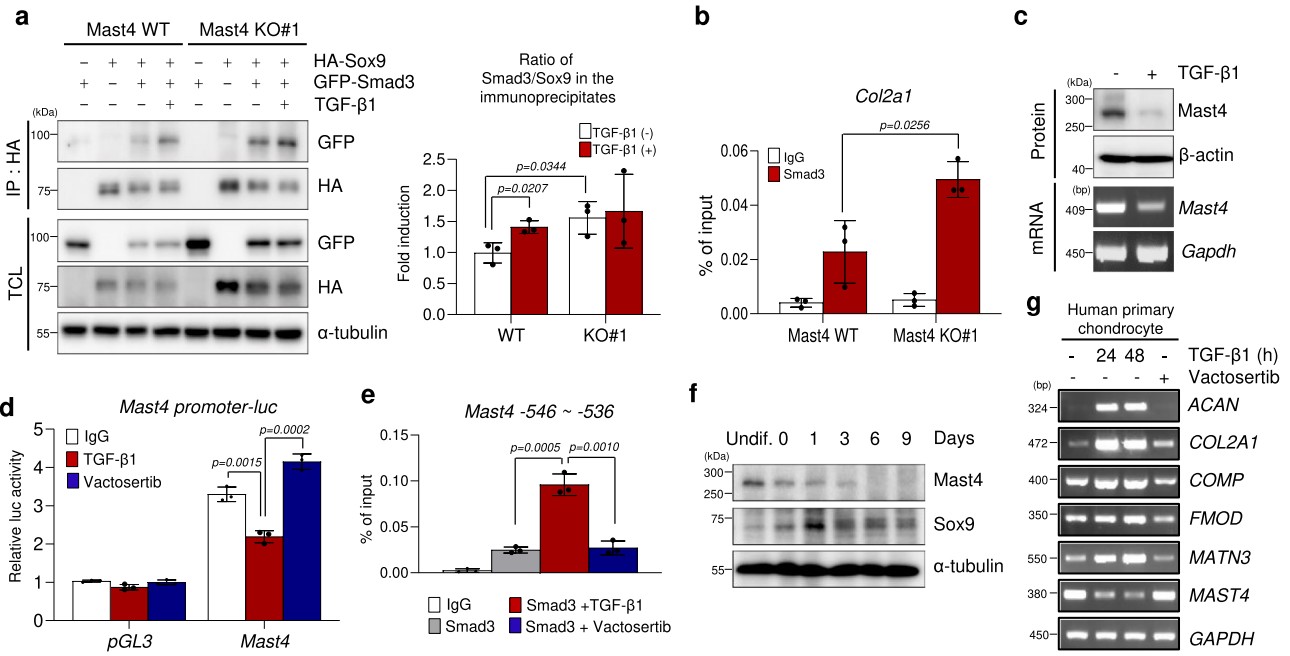

**Fig. 3 TGF-β1-induced suppression of Mast4 enhances chondrogenesis by increasing Sox9-Smad3 association. a** Sox9 and Smad3 were co-transfected to wild-type and Mast4-depleted C3H10T1/2 cells, followed by TGF-β1 treatment (5 ng/ml for 30 min). Sox9 was immunoprecipitated using HA antibody. Band intensities representing GFP-Smad3 and HA-Sox9 expression level in the immunoprecipitates were converted by densitometry using ImageJ software into the ratio of Smad3 to Sox9. Data are reported as mean ± SD of three independent experiments ($n = 3$). Unpaired two-tailed Student's t test ($P < 0.05$) was conducted for statistical analysis. **b** Smad3 ChIP on *Col2a1* gene assay was performed in differentiating C3H10T1/2 cells for 6 days. **c** Mast4 expression was examined by western blot and RT-PCR of C3H10T1/2 cells treated with TGF-β1 for 48 h. **d** Luciferase assay was conducted in undifferentiated C3H10T1/2 cells. Vactosertib (0.5 μM) was pre-treated for 2 h prior to TGF-β1 (5 ng/ml) treatment for 24 h. **e** Smad3 ChIP assay on *Mast4* gene was conducted in C3H10T1/2 cells undergoing chondrogenic differentiation for 6 days. TGF-β1 (5 ng/ml) and Vactosertib (0.5 μM) were treated for 48 h and 50 h, respectively, before harvest. **f** Endogenous Mast4 and Sox9 protein expression was examined by western blotting in differentiating C3H10T1/2 cells. **g** TGF-β1 (5 ng/ml) was treated for 24 h and 48 h, and Vactosertib (0.5 μM) was treated for 48 h to human primary chondrocytes. **b**, **d** and **e** Data are representative mean ± SD of three independent experiments, each conducted in triplicate ($n = 3$). Unpaired two-tailed Student's t test ($P < 0.05$) with Benjamini-Hochberg correction for multiple tests was conducted for all statistical analyses. **a**, **c**, **f** and **g** The representative results were obtained from at least three independent experiments. TCL: total cell lysates. Uncropped blots in **a**, **c**, **f** and **g** are shown in Source Data file. Source data for **a**, **b**, **d** and **e** are provided as a Source Data file.

of C3H10T1/2 cells (Supplementary Fig. 17). Taken together, our data suggest that Mast4, which is stabilized by Wnt-mediated GSK-3β inhibition, may function as an important mediator of Wnt/β-catenin-induced osteogenesis.

**Skeletal abnormalities in *Mast4*$^{-/-}$ mice.** To examine the role of Mast4 in MSC differentiation, we generated *Mast4*$^{-/-}$ mice using CRISPR/Cas9-mediated knockout system (Supplementary Fig. 18a–d). Even though *Mast4*$^{-/-}$ mice were viable, they became smaller in size, compared to the wild-type littermates (Supplementary Fig. 18e). Although we cannot exclude the possibility that Mast4 effects on other organs also contribute to the small size of the mice, we focused on investigating whether disruption of the *Mast4* gene affects differentiation of MSCs during endochondral ossification in the long bones since MSCs are found in many internal organs of adult mice. The expression of Mast4 was observed throughout the tibial growth plate of the *Mast4*$^{+/+}$ mice at postnatal (PN) 1 day (Supplementary Fig. 19a). Interestingly, increased expression of Mast4 was observed in hypertrophic chondrocytes, while a smaller portion was localized in proliferating chondrocytes in the tibias. This observation supports our findings regarding enhanced chondrogenic differentiation of C3H10T1/2 cells by Mast4 depletion and suggests a role of Mast4 in mediating a switch to bone growth in the terminal stage of chondrocyte differentiation. Indeed, stronger expression of Sox9

and Col2a1 proteins, two key markers of chondrocytes, was observed in the growth plates of *Mast4*$^{-/-}$ mice at PN 1 day (Figs. 5a, b). Pentachrome staining also displayed an increased hypertrophic layer in the proximal epiphysis of *Mast4*$^{-/-}$ mice at PN 1 day (Fig. 5c and Supplementary Fig. 19b, c). Furthermore, more yellowish pentachrome staining was detected in the growth plates of *Mast4*$^{-/-}$ mice at PN 3 weeks, suggesting that Mast4 depletion may contribute to collagen synthesis in the growth plate (Fig. 5d). The tibial growth plate thickness of *Mast4*$^{-/-}$ mice showed no significant difference between *Mast4*$^{+/+}$ mice at PN 1 day and 3 weeks, while being significantly reduced at PN 6 weeks (Supplementary Fig. 19b–d). However, the ratio of hypertrophic layer to the total thickness of the growth plate was significantly increased in *Mast4*$^{-/-}$ mice at PN 3 and 6 weeks (Supplementary Fig. 19e). This observation suggests that excessive cartilage synthesis in hypertrophic zone of the growth plate of *Mast4*$^{-/-}$ mice resulted in reduced proliferation of the growth plate, leading to abnormal ossification. Since the increased hypertrophic layer is the major phenotype observed in both *Mmp9*$^{-/-}$ and *Mmp13*$^{-/-}$ mice[26], increased hypertrophic layer in *Mast4*$^{-/-}$ mice may be associated with down-regulation of *Mmp9* and *Mmp13* observed in Mast4-depleted C3H10T1/2 cells (Fig. 1d and Supplementary Fig. 2c).

On the other hand, the μCT analyses of *Mast4*$^{-/-}$ mice demonstrated an osteoporotic phenotype with significantly reduced metaphyseal trabecular bones, more porous and

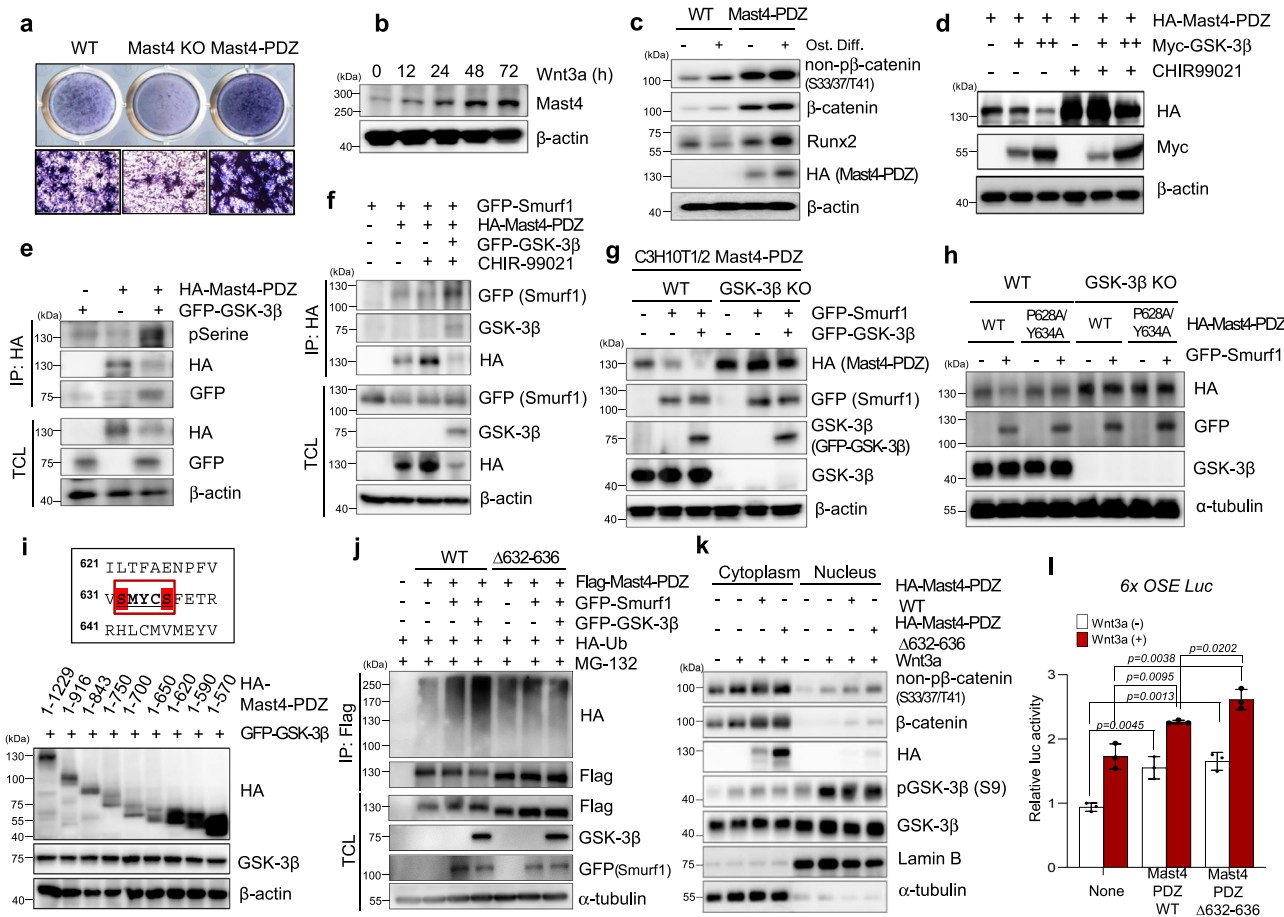

**Fig. 4 Wnt induces osteogenesis by enhancing Mast4 stability through inhibition of GSK-3β. a** Representative ALP staining results of osteogenic differentiated C3H10T1/2 cells obtained from at least three independent experiments. **b** MC3T3-E1 preosteoblasts were treated with Wnt3a conditioned medium for the indicated time. **c** Wild-type and Mast4-PDZ-overexpressing C3H10T1/2 cells were differentiated into osteoblasts for 10 days. **d** Mast4-PDZ and GSK-3β were transfected to C3H10T1/2 cells, followed by CHIR-99021 treatment (10 μM for 9 h). **e** Mast4-PDZ and GSK-3β were transfected to C3H10T1/2 cells, followed by immunoprecipitation assay. The bands, which were recognized by phosphoserine antibody, were later reprobed with HA antibody. **f** Smurf1, Mast4-PDZ and GSK-3β were transfected to C3H10T/12 cells treated with CHIR-99021 treatment (10 μM for 9 h), followed by immunoprecipitation assay. **g** Smurf1 and GSK-3β were transfected to wild-type and GSK-3β-depleted Mast4-PDZ-overexpressing C3H10T1/2 cells. **h** Mast4-PDZ WT, P628A/Y634A and Smurf1 were transfected to wild-type and GSK-3β-depleted C3H10T1/2 cells. **i** Various Mast4 kinase domain deletion mutants and GSK-3β were transfected to C3H10T1/2 cells. **j** Flag-Mast4-PDZ WT and Δ632-636 were co-transfected into C3H10T1/2 cells together with HA-Ubiquitin in the absence or presence of Smurf1 and GSK-3β. Cell lysates were immunoprecipitated with Flag antibody and immunoblotted with the indicated antibodies. **k** MC3T3-E1 cells were transfected with Mast4-PDZ WT and Δ632-636, followed by treatment of Wnt3a conditioned medium (2.5 h). **l** 6xOSE-Luc, Mast4-PDZ WT and Δ632-636 were transfected to C3H10T1/2 cells, followed by treatment of Wnt3a conditioned medium (18 h). Data are representative mean ± SD of three independent experiments, each conducted in triplicate ($n = 3$). Unpaired two-tailed Student's t test ($P < 0.05$) with Benjamini-Hochberg correction for multiple tests was conducted for all statistical analyses. **b–k** The representative results were obtained from at least three independent experiments. TCL: total cell lysates. Uncropped blots in **b–k** are shown in Source Data file. Source data for **l** are provided as a Source Data file.

thinner cortical bones, and decreased bone volume and mineral density (Fig. 5e and Supplementary Fig. 20a, b). Furthermore, the electron probe microanalyzer (EPMA) exhibited lower levels of critical mineral ions for bone development and bone health, such as magnesium (Mg), phosphate (P) and calcium (Ca), in the tibias of 6-week-old $Mast4^{-/-}$ mice (Supplementary Fig. 20c). In particular, we observed that $Mast4^{-/-}$ mice showed significantly decreased bone formation, measured by double calcein labeling, and shorter limb length compared to $Mast4^{+/+}$ mice (Fig. 5f and Supplementary Fig. 20d). Consistently, reduced expression of Osterix, an osteoblast marker crucial for bone formation, and Mmp13, a marker for mature osteoblasts and a target of Osterix, were observed in the proximal tibias and distal femurs of $Mast4^{-/-}$ mice at PN 1 day (Fig. 5g and Supplementary Fig. 21).

We further isolated skeletal stem cells, a purified population of CD45-TER119-TIE2-ITGAV+THY1-6C3-CD105-, from $Mast4^{+/+}$ and $Mast4^{-/-}$ mice using the expression of cell surface markers[27] (Supplementary Fig. 22). Functional assessment of these stem cells by in vitro colony formation assay revealed enhanced chondrogenic differentiation, shown by stronger alcian blue staining intensity, but suppressed osteogenic differentiation, shown by weaker Alizarin Red S staining intensity, abilities of the skeletal stem cells of $Mast4^{-/-}$ mice (Fig. 5h and Supplementary Fig. 23). We also isolated bone marrow-derived stem cells (BMSC) and induced chondrogenic and osteogenic differentiation. Consistently, enhanced chondrogenic but reduced osteogenic differentiation was observed in BMSCs of $Mast4^{-/-}$ mice (Supplementary Fig. 24). Collectively, our data suggest that Mast4 regulates the fate of MSC differentiation into cartilage or bone in vivo.

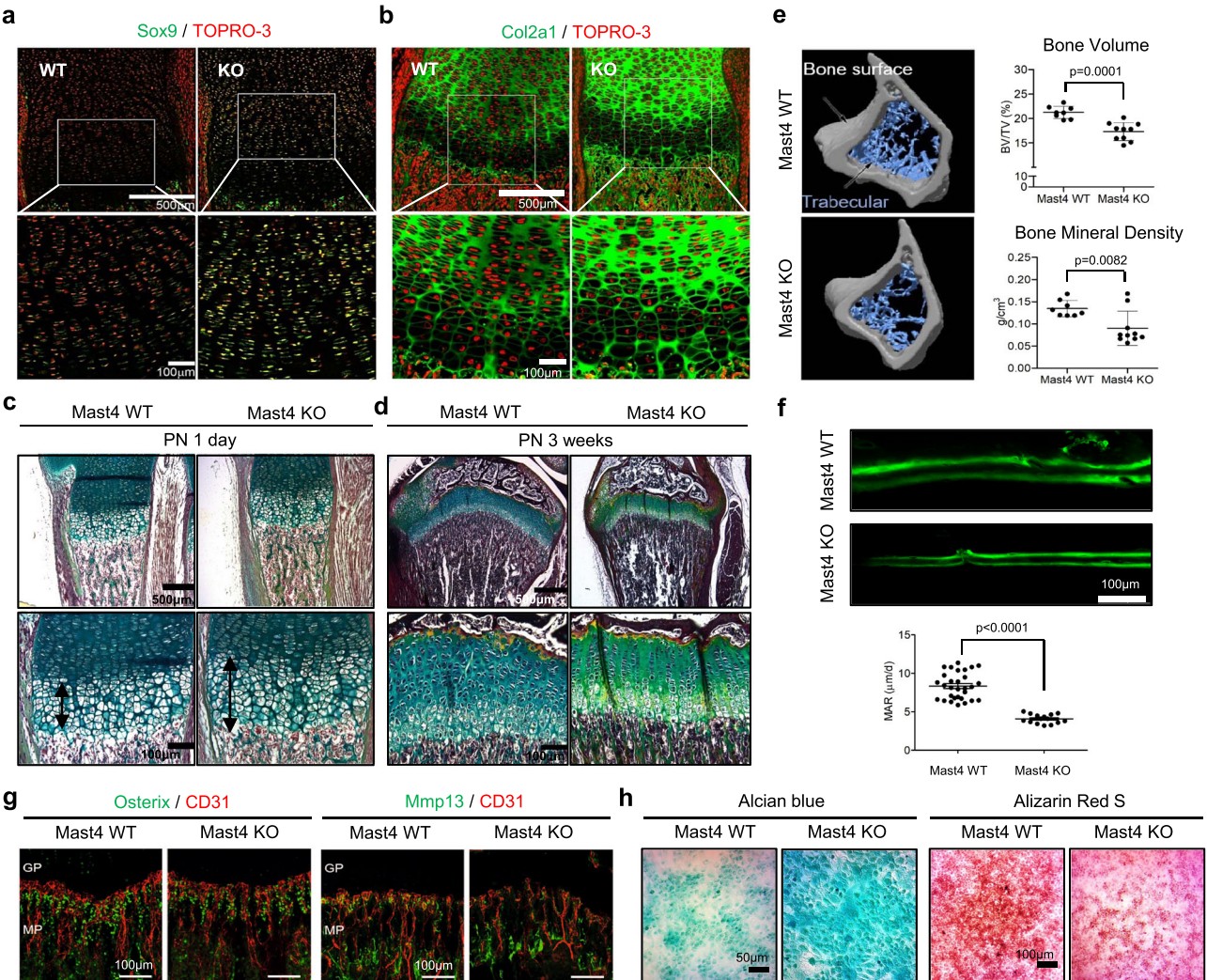

**Fig. 5 Mast4 depletion induces altered chondrogenesis and osteogenesis during development. a**, **b** Immunofluorescence images of Sox9 and Col2a1 in the growth plate of $Mast4^{+/+}$ and $Mast4^{-/-}$ mice at PN 1 day ($n = 3$). **c**, **d** Pentachrome staining of the tibial growth plates of the (**c**) 1-day-old and (**d**) 3-week-old $Mast4^{+/+}$ and $Mast4^{-/-}$ mice ($n = 5$). Black arrows indicate the hypertrophic zone. **e** The µCT images of the trabecular bone in the tibias of $Mast4^{+/+}$ and $Mast4^{-/-}$ mice. Total bone volume and bone mineral density (BMD) of trabecular bone were measured from the µCT images ($n = 8$ for $Mast4^{+/+}$ and 10 for $Mast4^{-/-}$ mice). **f** Bone formation was visualized by double calcein labelling at 7d and 2d prior to sacrifice, and the distance between labels were measured at 3 points per tibia ($n = 10$ tibias for $Mast4^{+/+}$ and 5 tibias $Mast4^{-/-}$ mice). MAR: mineral apposition rate. **g** Representative images of the immunofluorescence staining of Osterix and Mmp13 near metaphyseal blood vessels (CD31) in the proximal tibia of 6-week-old $Mast4^{+/+}$ and $Mast4^{-/-}$ mice ($n = 3$). GP: growth plate, MP: metaphysis. **h** Representative alcian blue and Alizarin Red S staining results obtained from in vitro colony formation assay using skeletal stem cells isolated from 5-week-old $Mast4^{+/+}$ and $Mast4^{-/-}$ mice ($n = 5$). **e**, **f**, Data are reported as mean ± SD. Unpaired two-tailed Student's t test ($P < 0.05$) was conducted for all statistical analyses. Source data for **e**, **f** are provided as a Source Data file.

**Phenotype of cartilage in the tibias of $Mast4^{-/-}$ mice**. To gain a better understanding of the role of Mast4 in MSC differentiation, we conducted RNA sequencing by collecting and combining RNAs obtained from bone and cartilage of the tibias of $Mast4^{-/-}$ mice at PN 1 day with those of wild-type mice (3 mice per each group). Differentially expression (DE) analysis exhibited tissue-specific expression with 175 up-regulated (CL1) and 181 down-regulated (CL2) genes in bone, and 108 up-regulated (CL4) and 327 down-regulated (CL5) genes in cartilage of $Mast4^{-/-}$ mice (Supplementary Fig. 25a). Gene ontology (GO) enrichment analysis revealed the high association with skeletal system development (D in Supplementary Fig. 25b; represented by CL2, CL3, and CL4; $P = 4.5×10^{-8}~9.3×10^{-4}$). Remarkably, 17 of the 120 common differentially expressed genes (DEGs; CL3), mostly identified as Sox9 target genes that are cartilage-specific for skeletal system development, showed distinct switches in expression between bone and cartilage (Supplementary Fig. 25b), suggesting the functional involvement in MSC differentiation. Gene set enrichment analysis for DEGs also revealed that significant numbers of DEGs in cartilage were assigned to cartilage development (FDR = 0.001), while DEGs in bone were assigned to skeletal system development (FDR = 0.004) and Wnt signaling pathway (FDR = 0.002) (Fig. 6a, b). Particularly, most of Sox9 target genes were up-regulated in cartilage of $Mast4^{-/-}$ mice, but not in bone. A network of Sox9 and Runx2 target genes showing differential expression in cartilage and/or bone of $Mast4^{-/-}$ mice were analyzed. These Sox9 and Runx2 targets were highly interacted with the genes related to skeletal system development, including cartilage and bone development, TGF-β signaling, BMP signaling, and Wnt signaling (Fig. 6c). Genes related to cartilage development were highly interacted with Sox9 target genes (i.e. Tgfb1, Bmp7 and Bgn; DE in cartilage). However, genes related to

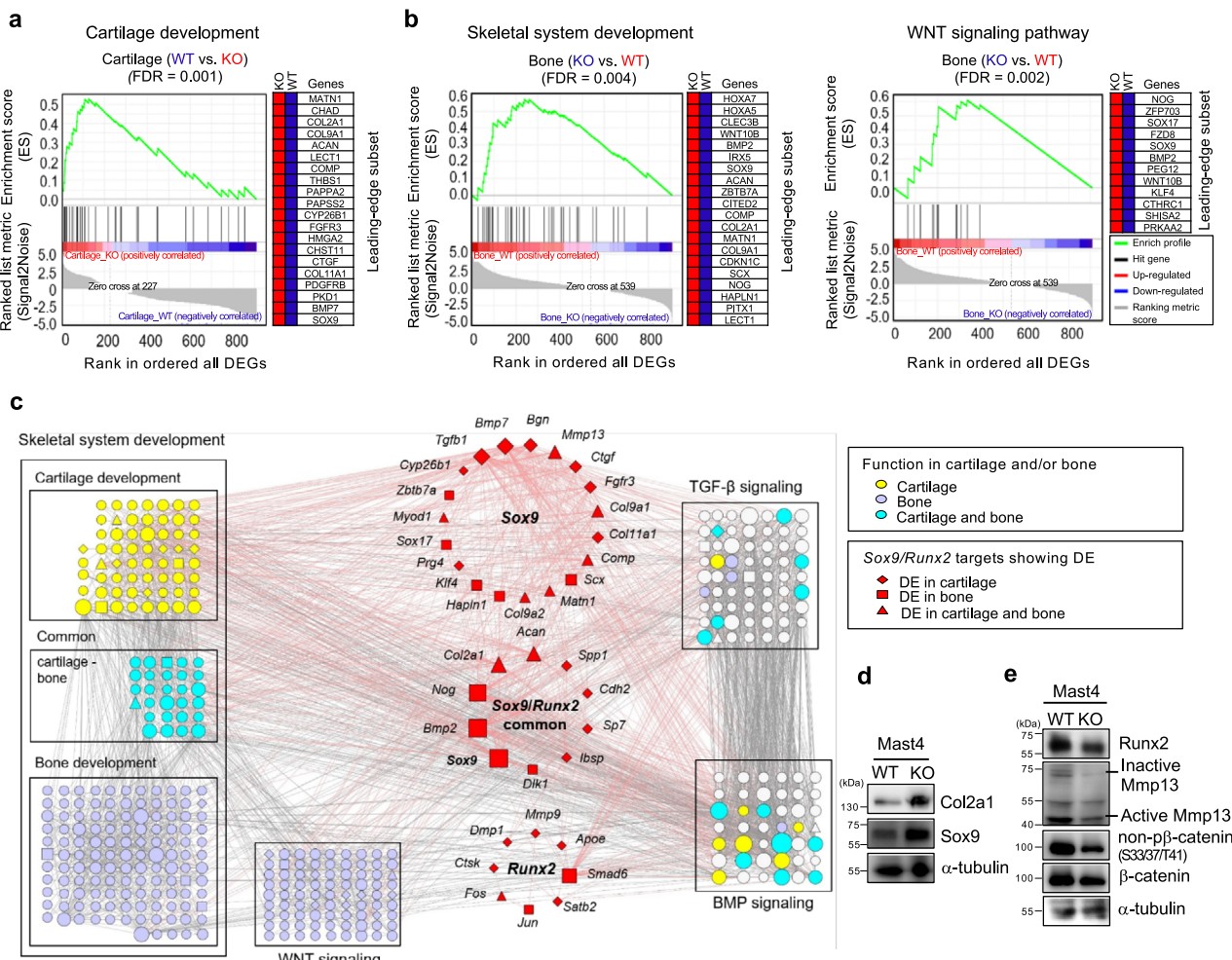

**Fig. 6 Identification of the genes regulated by Mast4 in mouse cartilage and bone. a, b** RNA sequencing was conducted by collecting and combining RNAs obtained from cartilage and bone of the tibias of $Mast4^{+/+}$ and $Mast4^{-/-}$ mice at PN 1 day (3 mice per each group). **a** The enrichment of upregulated genes associated with cartilage development in the cartilage of $Mast4^{-/-}$ mice. Twenty up-regulated genes were predicted as the leading-edge subset of the enriched gene set. **b** The enrichment of upregulated genes associated with skeletal system development and Wnt signaling pathway in the bone of $Mast4^{+/+}$ mice. In the plot, 20 and 12 genes were upregulated as the leading-edge subsets of the enriched gene sets in $Mast4^{+/+}$ mice. Gene set enrichment analysis was applied with a background dataset consisting of all DEGs analyzed from cartilage and bone of the tibias of $Mast4^{+/+}$ and $Mast4^{-/-}$ mice. **c** Mast4-regulated transcriptional network of Sox9 and Runx2 targets showing differential expression in cartilage and bone. The Sox9 and Runx2 targets are linked to the genes related to the functions of skeletal system development, including cartilage and bone development, TGF-β signaling, BMP signaling and Wnt signaling. **d** Western blot analysis of Col2a1 and Sox9 proteins in the cartilage tissues of $Mast4^{+/+}$ and $Mast4^{-/-}$ mice at PN 1 day. **e** Western blot analysis of Runx2, Mmp13 and β-catenin proteins in the bone tissues of $Mast4^{+/+}$ and $Mast4^{-/-}$ mice at PN 1 day. **d, e** The representative results were obtained from at least three independent experiments. Uncropped blots in **d, e** are shown in Source Data file.

bone development and Wnt signaling were mostly interacted with the common target genes of *Sox9* and *Runx2* (i.e. *Nog*, *Bmp2* and *Sox9*; DE in bone), or *Runx2* targets (i.e. *Smad6*; DE in bone). Interestingly, the network of *Sox9* target genes showed the regulation of collagen family and CL3-switch genes in bone and cartilage. Collectively, our results suggest that Mast4 is a key regulator of a transcriptional network associated with skeletal system development.

Next, expression of the selected cartilage matrix genes and Sox9 target genes in the transcriptional network was further examined in cartilage tissues isolated from the tibias of wild-type and $Mast4^{-/-}$ mice at PN 1 day by qRT-PCR (Supplementary Fig. 26a, b). Particularly, collagenase genes (*Mmp9* and *Mmp13*) were down-regulated, supporting our observation of increased hypertrophic layer in $Mast4^{-/-}$ mice (Fig. 5c). We also found that mRNA expression of the genes associated with osteoblast differentiation was decreased in the bone tissues of $Mast4^{-/-}$ mice (Supplementary

Fig. 26c, d). Consistent with these observations, the increase of Sox9 and Col2a1 and the decrease of β-catenin, Runx2, and Mmp13 protein expressions were observed in the cartilage and bone tissues, respectively, of $Mast4^{-/-}$ mice (Fig. 6d, e). Taken together, our results of the molecular phenotype of $Mast4^{-/-}$ mice support that Mast4 regulates the expression of the aforementioned genes through modulation of Sox9, ultimately facilitating chondrogenesis in vivo.

**Mast4 depletion facilitates cartilage formation and repair.** To further investigate the effect of Mast4 depletion on chondrogenesis in vivo, differentiated wild-type and Mast4-depleted C3H10T1/2 micromass cultures were subcutaneously implanted into nude mice for assessment of cartilage formation (Supplementary Fig. 27a). After 2 weeks, the grafted tissues formed by Mast4-depleted cells showed a significant increase in volume (Fig. 7a). Previous reports showed that cartilage formation from

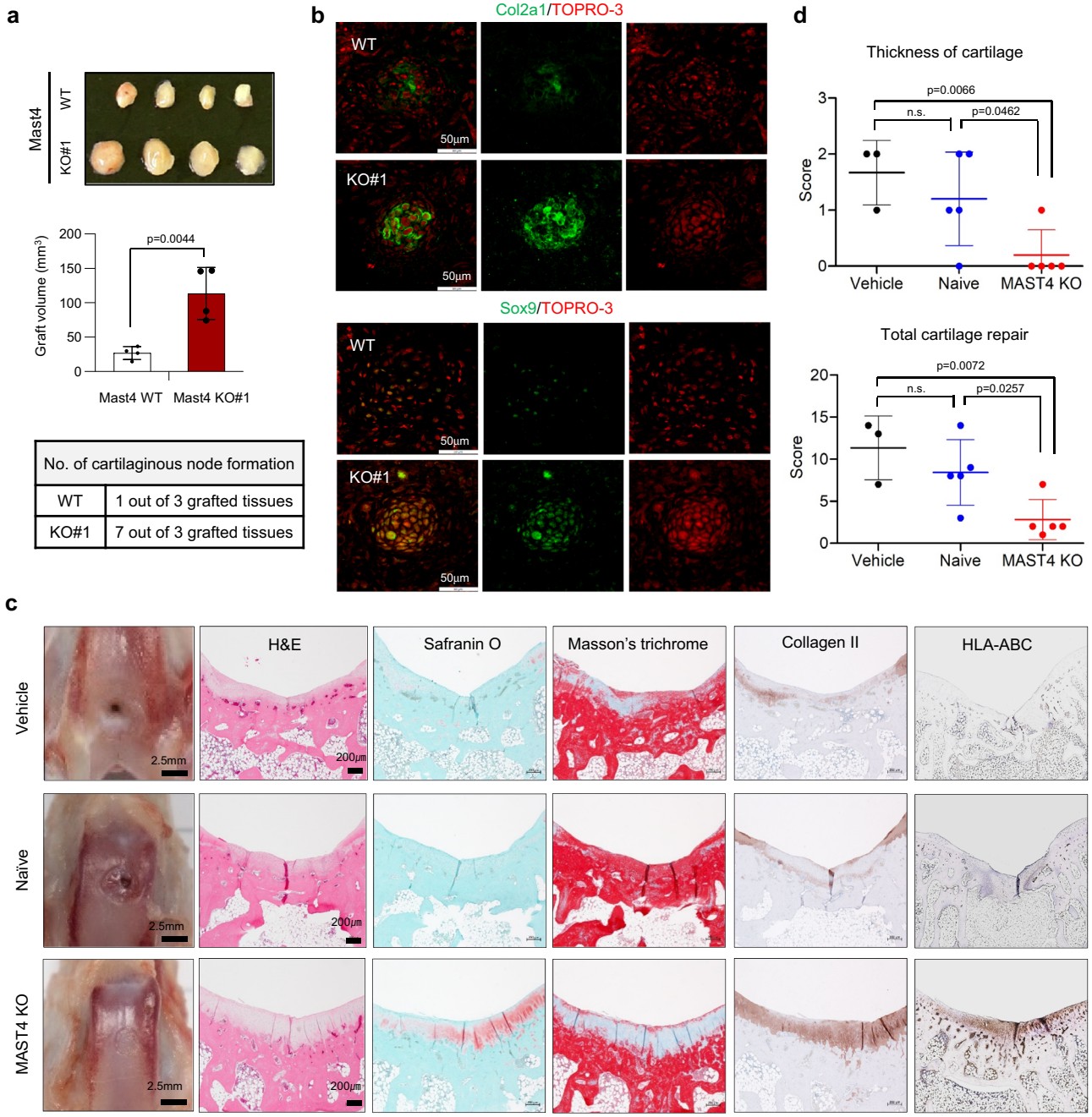

**Fig. 7 Transplantation of Mast4-depleted MSCs improves cartilage formation and repair in vivo. a** A hundred micromass cultures of the wild-type and Mast4-depleted C3H10T1/2 cells cultured in a 150 mm dish with chondrogenic differentiation medium for 4 days were subcutaneously injected into each side of the flanks of athymic nude mice (n = 4; WT on the left flank and KO on the right flank). Ectopic masses were retrieved at 2 weeks after implantation. The values given are mean ± SD of the volume of grafts from 4 mice. Unpaired two-tailed Student's t test (P < 0.05) was conducted for statistical analysis. **b** Multiple cartilage structures found in the grafts of wild-type and Mast4-depleted C3H10T1/2 cells were confirmed by immunofluorescence staining of Col2a1 and Sox9. Merge of Sox9 and TOPRO-3 is shown in yellow. The representative images were obtained from immunostainings of 1 cartilaginous node formed by wild-type C3H10T1/2 cells and 7 different cartilaginous nodes formed by Mast4-depleted C3H10T1/2 cells. **c** Gross and microscopic appearances of full-thickness cartilage defects in the trochlear groove in a rabbit model at 12 weeks post-transplantation. Vehicle: PBS treatment without hBMSC transplantation. **d** Wakitani cartilage repair scores of the regenerated cartilage. Data are represented as mean ± SD. **c**, **d** n = 3 for vehicle-treated group, n = 5 for naïve and MAST4-depleted hBMSC-transplanted groups. One-way ANOVA with Dunnett's correction for multiple comparisons was conducted for statistical analyses for **d**. Source data for **a**, **d** are provided as a Source Data file.

implantation of MSCs was barely observed by subcutaneous implantation, while it was more commonly observed by using diffusion chamber[28] or implanting inside a cartilaginous defects[29]. Similar to these reports, the grafts formed by both wild-type and Mast4-depleted C3H10T1/2 cells were not comprised of high portion of cartilaginous nodes. However, we found an increased incidence of cartilaginous node formation in the grafts formed by Mast4-depleted cells. To confirm whether the nodes were comprised of cartilage, tissue sections were stained with pentachrome (Supplementary Fig. 27b). An increased collagen deposition around the nodes was shown in the grafts formed by Mast4-depleted cells. Moreover, Col2a1, a chondrogenic marker

protein, and Sox9 proteins were highly expressed in the grafts formed by Mast4-depleted cells (Fig. 7b). Taken together, these results imply that depletion of Mast4 may enhance chondrogenic commitment of MSCs in vivo.

Next, we assessed the effect of MAST4 depletion in human bone marrow-derived stem cells (hBMSC) on cartilage repair in a rabbit full-thickness cartilage defect model. Since hBMSC were unable to form colonies from individual cells, the pools of CRISPR/Cas9-mediated MAST4-depleted cells showing at least 70% of indel, frameshift or 21+ bp indel were used. No abnormal findings or severe inflammatory reactions were observed. While the defects in the knees treated with vehicle (PBS) or transplanted with naïve hBMSCs were vacant and distinguishable from the surrounding tissues, the knees transplanted with MAST4-depleted hBMSCs exhibited smooth white repaired tissues which covered the defect without showing obvious margin with the normal surrounding cartilage (Fig. 7c and Supplementary Fig. 28). The architecture of the repaired tissues in the MAST4-depleted hBMSCs-transplanted knees resembled those of normal surrounding cartilage, whereas those in the naïve hBMSCs-transplanted knees showed irregular surface and noticeable gaps in the border area. In addition, increased Safranin O and Massons' trichrome staining were noted in the repaired tissues of MAST4-depleted hBMSCs-transplanted knees without significant gaps, indicating enhanced production of cartilage tissues and collagen matrix, respectively. Immunohistochemical staining against type II collagen further revealed significantly strong expression and similar density with the surrounding normal tissues by transplantation of MAST4-depleted hBMSCs. Overall, the modified Wakitani score analysis[30] demonstrated that the transplantation of MAST4-depleted hBMSCs to the defect sites significantly improved cartilage repair and regeneration (Fig. 7d and Supplementary Fig. 29). Collectively, these results suggest that Mast4 depletion may be a useful strategy for MSC-based therapy for cartilage regeneration.

## Discussion

Although MSCs are intensively researched with the aim to be used in regenerative therapy, the molecular mechanisms governing the differentiation of MSCs are not fully understood. Our results suggest that Mast4 is a key molecule that determines the commitment and differentiation of MSCs towards a chondrogenic or osteogenic cell fate. We have demonstrated that TGF-β1-mediated suppression of Mast4 gene transcription leads to the increase of Sox9 protein and Smad3-Sox9 association, which results in increased Sox9 transcriptional activity, ultimately initiating MSCs to favor chondrogenesis at the expense of bone formation. In regard to osteogenesis, we have shown that Wnt-mediated inhibition of Mast4 protein degradation by inhibiting GSK-3β activity leads to the increase of β-catenin protein and Runx2 transcriptional activity, ultimately initiating MSCs to favor osteogenesis (Supplementary Fig. 30).

The context-dependent nature of TGF-β has been delineated throughout the decades. Particularly, the cytostatic effect of TGF-β has shown to be orchestrated by transcriptional activation of CDK inhibitors and repression of *c-Myc*, highlighting its roles in the treatment of cancers[31]. Numerous studies have also identified the function of TGF-β in determining the fate of multipotent stem cells during developmental processes. In regard to endochondral ossification during skeletal development, TGF-β promotes mesenchymal condensation and chondrogenesis, but inhibits chondrocyte maturation and differentiation into osteocytes, indicating its sequential regulation along specific lineages[32]. The bi-functionality of TGF-β signal during skeletal development are supported by observations in animal models[15,33]. We observed suppression of Mast4 by TGF-β during chondrogenesis in vitro and predominant expression of Mast4 in hypertrophic chondrocytes in vivo. These observations speculate that TGF-β exerts diverse regulatory influences on skeletogenesis through specific regulation of Mast4 at different stages. Moreover, our observation of Mast4 exerting no influence on the cytostatic effect of TGF-β provides compelling evidence in favor of Mast4 being a critical mediator of the TGF-β-induced chondrogenic differentiation of MSCs.

Various transcription co-factors serve as Smad partners aiding in target gene recognition and transcriptional regulation[34]. Regarding Smad-mediated gene repression, Smad3 inhibits Runx2 activity through direct interaction, ultimately diminishing osteoblast differentiation[35]. E2F4/5 has been demonstrated as a co-repressor in TGF-β-induced repression of *c-Myc*[36]. TGF-β-induced *SpB* repression is associated with Smad3 interaction with Nkx2.1[37]. Interestingly, the expected binding sites of E2F4 and Nkx2.1 near the Smad3-binding site were recognized through analysis of the *Mast4* promoter region. Thus, it would also be important to investigate whether these co-transcription factors are involved in TGF-β/Smad3-mediated Mast4 regulation. In addition, discovery of novel co-transcription factors that regulate *Mast4* expression along sequential stages of chondro-/osteogenic differentiation would benefit the understanding of cartilage and bone development and their regulation.

Post-translational modifications (PTMs) modulate protein functions and stability, and fine tune signal transduction[38]. Here, we demonstrated the impacts of PTMs on the TGF-β1-Mast4-Sox9 axis during chondrogenesis. A number of signaling pathways and PTMs have been exhibited to regulate Sox9, a master transcription factor during chondrocyte differentiation, by controlling a repertoire of cartilage-related ECM genes at the early stage[10,16,39,40]. Our observations illustrate that Mast4 promotes Sox9 degradation by inducing Sox9 phosphorylation at serine 494. Even though it remains to be elucidated whether Mast4-induced Sox9 phosphorylation is recognized by any of the E3 ligases for subsequent Sox9 and Mast4 degradation, our study shows that Mast4 is likely to be an important factor in controlling Sox9 activity. E6-AP/UBEA is an E3 ligase that induces ubiquitin-mediated proteasomal degradation of Sox9 in hypertrophic chondrocytes during endochondral ossification[41]. Considering that Mast4 is predominantly expressed in hypertrophic chondrocytes, it may be worth examining whether Mast4 co-operates with E6-AP/UBEA to regulate Sox9 stability in hypertrophic chondrocytes. Moreover, examination of the regulation of Mast4 on Sox9 stability through phosphorylation at serine 494 in vivo and subsequent chondrogenic differentiation ability may be necessary in the future study. In addition, a previous study indicated the importance of the delicate balance of Sox9 activity in MSC for proper differentiation. Our observation of chondrocyte accumulation in the most terminally differentiated hypertrophic state and their delayed exit from the growth plate, shown in the endochondral bones of *Mast4$^{-/-}$* mice, may be explained by Mast4 deficiency-mediated overexpression of Sox9 and collagen as well as reduction of Mmp9 and Mmp13.

With regard to Wnt/β-catenin, different mechanisms have been reported to explain Wnt-mediated β-catenin stabilization[42]. Notably, Wnt inhibits GSK3 activity towards β-catenin in various ways. Given that phosphorylation by GSK3 often marks the target proteins for ubiquitination and proteolysis, our findings that inhibition of Mast4 phosphorylation by GSK-3β increases the stability of Mast4 and subsequent β-catenin reinforce the action of Wnt/β-catenin signaling in MSCs selecting osteoblastic fate[24,43]. Furthermore, it would be worth examining Mast4 protein level in GSK-3β-deficient mice and GSK-3β inhibitor-administered mice that display increased bone formation and bone mass[21,44,45].

Mast4 belongs to the MAST kinase family, consisting of Mast1-4 and Mastl[46]. Mast1 through 4 share a similar domain organization having a kinase domain, a PDZ domain, and a domain of unknown function (DUF). Currently, little is known about the biological roles of the MAST kinase family. Several studies have demonstrated the association of Mast 1, 2, and 3 with cancers[47,48]. Besides its role as a neuroprotective mediator[49–51], Mast4 has been reported to undergo O-GlcNAc modification, of which global elevation is frequently observed during osteoblast differentiation[52]. In addition, it was demonstrated that Mast4 mediated FGF-2 signaling, known to play a role in bone formation, in Sertoli cells through induction of ERM phosphorylation at serine 367 residue[53]. Meanwhile, the microtubule cytoskeletons have shown to contribute to the osteogenic differentiation of MSCs[54]. Since the MAST kinase family shares a high degree of similarity in protein domains that are considered as structural and functional building blocks, it is likely that the MAST kinase family members are critical cellular mediators of a variety of signal transduction in normal and diseased states. In addition, examination of Mast4 regulation in the differentiation of MSCs into various lineages, including osteoblasts or adipocytes, may be worth further investigation.

In conclusion, we have demonstrated that Mast4 is a crucial mediator in MSC commitment towards chondro-osteogenic differentiation pathway. Our findings implicate a function of Masts4 in the limiting of Sox9 transcriptional activity to determine the fate of MSC development into cartilage or bone. Therefore, in the context of cell therapy, Mast4 will be an ideal target for potential MSC therapy.

## Methods

All animal studies were approved by the Institutional Animal Care and Use Committee of KNOTUS Co., Ltd and Institutional Animal Care and Use Committee of Center for Phenogenomics Animal Research Facility and Woojung BSC, and performed in accordance with ethical and procedural guidelines. The laboratory mice were maintained on a 12-h light/dark cycle at room temperature (20-22 °C) with constant humidity (40 ± 10%).

**Generation of Mast4 knockout mice by CRISPR/Cas9 technology.** For generation of Mast4 knockout mice by CRISPR/Cas9-mediated gene targeting, we targeted exon 1 and exon 15 of *Mast4* (RefSeq Accession number: 175171): 5'-GGAAACTCTGTCGGAGGAAG-3' (exon 1) and 5'-GGCACAAA-GAGTCCCGCCAG-3' (exon 15). We then inserted each sequence into *pX330* plasmid, which carried both guide RNA and Cas9 expression units, received from Dr. Feng Zhang (Addgene, #42230)[55]. We named these vectors *pX330-Mast4-E1* and *pX330-Mast4-E15*.

The pregnant mare serum gonadotropin (5units) and the human chorionic gonadotropin (5 units) were intraperitoneally injected at a 48 h interval into female C57BL/6 J mice (Charles River Laboratories, Kanagawa, Japan), which were then mated with male C57BL/6 J mice. The *pX330-Mast4-E1* and *pX330-Mast4-E15* (circular, 5 ng/μl each) were co-microinjected into 231 zygotes collected from the oviducts of the mated female mice. The survived 225 injected zygotes were transferred into the oviducts in pseudopregnant ICR female, and 47 newborns were obtained. We collected genomic DNA from the tails of 31 founder mice that survived.

To confirm indel mutations induced by CRISPR/Cas9, we amplified genomic region including the target sites by PCR with the primers for exon 1 target (MAST4-1 genotype F: 5'-GTAGGGACTCCACGCTCCAG-3'; MAST4-1 genotype R: 5'-CCGGACCCTAGTCTCTTCG-3') and for exon 15 target (MAST4-15 genotype F: 5'-GGGTTCTCTGCGAAAGTCAG-3'; MAST4-15 genotype R: 5'-ATCCCTGTGTTCCGTTTCAG-3'). The PCR products were sequenced by using BigDye Terminator v3.1 Cycle Sequencing Kit (Thermo Fisher Scientific), MAST4-1 genotype F primer, and MAST4-15 genotype F primer. In male founder #38, we found indel mutations in both exon 1 and exon 15 without *pX330* random integration. To identify the indel sequence and whether indel mutations in exon 1 and exon 15 occurred on the same chromosome (cis manner), the founder #38 was mated with wild-type female, and the indel mutations in F1 were sequenced. We obtained 17 F1 newborns, and 12 of them carried 71 bp deletion (chr13:103,333,981-103,334,051: GRCm38/mm10) in exon 1 and 3 bp deletion (chr13:102,774,360-102,774,362) in exon 15 in a cis manner.

**CRISPR/Cas9-mediated deletion of the *Mast4* in C3H10T1/2 and human bone marrow-derived stem cells.** For C3H10T1/2 cells, lentiCRISPRv2 vector

(Addgene, #52961) was digested with BsmBI and ligated with annealed oligonu-cleotide targeting *Mast4* exon 1, 5'-TACCCTGCCGCTGCCGCACC-3' (Lenti-CRISPRv2-Mast4 Ex1) and exon 2, 5'- AGCAACCCAGATGTGGCCTG-3' (LentiCRISPRv2-Mast4 Ex2). To generate lentivirus, HEK293T cells were trans-fected with LentiCRISPRv2-Mast4 Ex1 and packaging vectors (pVSVG and psPAX2, Addgene #8454, #12260) using polyethylenimine at 70% confluency. Viral supernatant was harvested at 48 h post-transfection, filtered through 0.45-μm filters and applied to C3H10T1/2 cells. After puromycin-mediated selection, single-cell clones were grown in 96-well plates. From genomic DNA, the exon 1 and exon 2 regions of the *Mast4* gene were amplified using AccuPower™ PCR premix (Bio-neer). The indel mutations causing frameshift-mediated depletion of Mast4 protein were confirmed by sequencing. For human bone marrow-derived stem cells (hBMSC), we generated guide RNA (gRNA) using GeneArt™ Precision gRNA Synthesis Kit (Invitrogen) according to the manufacturer's protocol. Human bone marrow-derived stem cells at passage 5-6 were transfected with the gRNA targeting exon 5 (Forward: 5'-TAATACGACTCACTATAGAGCAACCGGAAAAGCTT AAT-3'; Reverse: 5'-TTCTAGCTCTAAAACATTAAGCTTTTCCGGTTGCT-3') and Cas9 protein (Toolgen) using the Neon Transfection System following the manufacturer's protocol. Since hBMSC were unable to form colonies from indi-vidual cells, the pools of edited cells were used for further chondrogenic differ-entiation, protein and mRNA isolation. The CRISPR/Cas9-mediated *Mast4* gene knockout efficiency in hBMSC was determined by ICE knockout analysis (www. synthego.com). Mast4-depleted hBMSC obtained >70 of ICE and KO scores, which indicates indel percentage and the proportion of cells having frameshift or 21+ bp indel, respectively, were used.

**Lentiviral shRNA Production/Infections.** Two different shRNAs targeting exon 15 and exon 22 (shMast4 Exon 15 F: CCGGCCCAGTT GATATGGCCAGAA TCTCGAGATTCTGGCCATATCAACTGGGTTTTTG, shMast4 Exon 22 F:CCGGCCGAAGTTTCTCCTGCTTAAACTCGAGTTTAAGCAGGAGA AACTTCGG TTTTTG) of Mast4 were designed, and annealed oligos were inserted into the pLKO.1 vectors. To generate shRNA lentivirus, 293 T cells were trans-fected with pLKO-shMast4 (#1 and #2) or scrambled control pLKO-pGL2 together with lentiviral packaging plasmids, psPax2 and VSV-G. At 48 h post transfection, viral supernatants were harvested and filtrated. C3H10T1/2 cells were infected with shRNA lentivirus and polybrene (8 μg/ml) for 24 h, followed by puromycin selection (4 μg/ml).

**Cell culture and chondrogenic/osteogenic differentiation.** C3H10T1/2 cells (Clone 8, CCL-2260, ATCC), mouse bone marrow-derived mesenchymal stromal cells (mBMSC), and human embryonic kidney cell line HEK293T (CRL-3216, ATCC), were grown in Dulbecco's Modified Eagle's Medium (DMEM; LM001-05, WELGENE) containing 10% fetal bovine serum (FBS; S001-01, WELGENE) and 1% penicillin-streptomycin (P/S; LS202-02, WELGENE). ATDC5 (RCB0565, RIKEN BRC) cells were grown in DMEM/F-12 (11320033, Gibco) containing 5% FBS and 1% P/S. The hBMSC were kindly provided from SCM Lifescience (Incheon, S.Korea), where established hBMSC lines through the subfractionation culturing method[56]. Briefly, human bone marrow aspirates from the iliac crest of three healthy donors after written informed consent approved by Inha University Hospital Institutional Review Board; IRB number 10-51, were mixed with isolation medium and incubated. The supernatants containing floating bone marrow cells without the cells settled down to the bottom were repeatedly transferred to new 100-mm dishes. After 10-14 days of incubation, well-separated colonies were iso-lated, expanded and characterized. These were grown in DMEM (low glucose; LM001-11, WELGENE) containing 10% FBS and 1% P/S. The human primary chondrocytes, which were collected by straining collagenase-treated cartilage tis-sues obtained from 1-year-old human female donor[57], were also kindly provided by SCM Lifescience. These were grown in DMEM (17-205-CVR, Corning) containing 10% FBS (26140-079, Gibco), 20 mM L-glutamine (25030-081, Gibco), and 10 μg/ ml Gentamicin (15700-060, Thermo fisher). MC3T3-E1 cells (Subclone 4, CRL-2593, ATCC) were grown in Alpha Minimum Essential Medium (α-MEM) without ascorbic acid (LM008-53, WELGENE) containing 10% FBS and 1% P/S. All cells were cultured at 37 °C in a humidified 5% $CO_2$ incubator. For the micromass culture of C3H10T1/2 cells, $1 \times 10^5$ cells in a 10 μl drop of normal growth medium were seeded onto the culture dish, followed by an 2 h attachment period. Then, BMP-2 (150 ng/ml; PeproTech)-containing medium was added to the dish, and the medium was replaced every 48-72 h. For the pellet culture of hBMSCs, $2 \times 10^5$ cells were seeded onto a 15 ml conical tube and were grown in α-MEM containing 1% P/ S, $10^{-7}$M of dexamethasone (Sigma Aldrich), 1/100 of ITS + Premix Universal Culture Supplement (Corning), 50 ng/ml of ascorbic acid (Sigma Aldrich), 10 ng/ ml of TGF-β1 and TGF-β3 (R&D Systems), and 40 ng/ml of L-Proline (Sigma Aldrich) for 21 days. The medium was replaced every 48-72 h. For mBMSCs, cells were isolated from an aspirate of bone marrow harvested from the tibia marrow compartments and were cultured in DMEM containing 10% FBS for 3 h. Non-adherent cells were carefully removed, and fresh medium was resupplied. The cultured BMMSCs were differentiated to chondrocytes using the StemPro Chon-drogenesis Differentiation Kit (Thermo Fisher Scientific) according to the manu-facturer's instructions. For osteogenic differentiation of C3H10T1/2 cells, confluent cells were cultured in the maintenance medium supplemented with 50 μg/ml of

ascorbic acid (Sigma Aldrich), 10 mM of β-glycerophosphate (Sigma Aldrich), and 200 ng/ml of BMP-2 for 10 days. The medium was replaced every 48-72 h.

**Alcian blue and alkaline phosphatase (ALP) staining**. The differentiated cells were washed with PBS twice and fixed in 4% paraformaldehyde at room temperature for 5-10 min. The chondrogenic differentiated cells were stained with alcian blue solution (1% alcian blue in 0.1 M HCl, pH 1.0; Sigma Aldrich) overnight, followed by one wash with 0.1 M HCl and two with PBS. The osteogenic differentiated cells were stained with 5-bromo-4-chloro-3-indolyl-phosphate/nitro-blue tetrazolium solution (BCIP/NBT; Merck) for 30 min at 37 °C.

**The 3D spheroid formation assay**. The 3D spheroid formation of C3H10T1/2 cells using low-binding plate was conducted as previously reported[23]. Briefly, the round bottom ultra-low attachment 96-well microplate (Corning) was coated with gelatin (0.1%; Sigma Aldrich). Then, $1 \times 10^5$ cells in 50 µl of BMP-2 (150 ng/ml)-containing medium were added to each well of the coated microplate and cultured for 8 days. The medium was replaced every 48-72 h.

**RNA-Seq and bioinformatics analyses**. For the sample preparation of RNA sequencing using the differentiating C3H10T1/2 cells, total 30 high-density micromass cultures obtained from three separate induction of chondrogenic differentiation (10 masses/each induction) of the wild-type and Mast4-depleted (KO#1) C3H10T1/2 cells were combined together for RNA sequencing. For the RNA sequencing using the cartilage and bone of mice, cartilage and bone was dissected as follows. After euthanizing Mast4$^{+/+}$ and Mast4$^{-/-}$ mice at PN 1 day in a $CO_2$ chamber, the middle part of the femur was cut. After removing the skin, all the muscles were removed with forceps. The fibula was removed after amputation at the articular cartilage of the knee and ankle joint of tibia. The epiphysis was separated from the body of tibia along the boundary of the calcified zone with a 30 G needle. The separated epiphyseal cartilage and tibia bone were placed in Trizol® (Invitrogen). The tibia was hemisectioned and chopped with a razor blade in Trizol®. Each sample was homogenized with an equal amount of 0.5 mm stainless steel beads. Then, RNA was obtained through layer separation using chloroform and precipitation using isopropanol. Since the total amount of RNA obtained from each mouse at PN 1 day was not sufficient for RNA sequencing analysis, RNAs obtained from the cartilage or bone of the tibias of Mast4$^{+/+}$ and Mast4$^{-/-}$ mice at PN 1 day were combined (n = 3 per each group). RNA-Seq libraries were prepared using TruSeq RNA Sample Prep Kit according to the manufacturer's manual (Illumina, Inc., San Diego, CA) using 1 µg of the qualified RNA in each sample. After qPCR validation, libraries were subjected to paired-end sequencing with a 100 bp read length using an Illumina HiSeq 2500 platform, yielding an average of 57.7 million reads per library. The quality of raw reads was assessed with FastQC (version 0.11.9). Clean reads for each sample, in which average quality scores were greater than Q30, were aligned to the mouse reference genome GRCm38.p4mm10 using TopHat[58] with a set of gene model annotation. Gene expression was calculated as FPKM using Cufflinks. Differential expression analysis between the wild-type and Mast4-depleted samples was performed by using Cuffdiff[59] with a cutoff set at $P < 0.05$ and $\geq 1.5$-fold change in reference to qPCR validation of Sox9-targeted genes. Gene ontology (GO) enrichment analysis for DEG datasets was performed by DAVID[60] with a cutoff of $P < 0.001$., Interaction for genes related to cartilage and/or bone development, BMP signaling, TGF-β signaling, and Wnt signaling was searched using STRING database (https://string-db.org/) with high confidence score (≥0.7) and further analyzed using Cytoscape (www.cytoscape.org) on the basis of the degree of connectivity of the nodes. Gene Set Enrichment Analysis (GSEA) (www.gsea-msigdb.org/gsea/index.jsp)[61] was applied with a background dataset consisting of all DEGs analyzed from cartilage and bone of the tibias of Mast4$^{+/+}$ and Mast4$^{-/-}$ mice at PN 1 day that were expressed >0.3 FPKM, which balances the numbers of false positives and false negatives[62], in either Mast4$^{+/+}$ or Mast4$^{-/-}$ mice.

**Reverse transcription PCR (RT-PCR) and real-time RT-PCR**. Total RNA was prepared using EasyBlue (Boca Scientific). 2 µg of RNA was reverse-transcribed using M-MLV Reverse Transcriptase (Promega) according to the manufacturer's instructions. RT-PCR was conducted using AccuPower™ PCR premix (Bioneer) with specific primer pairs. Quantitative real-time PCR was performed using Power SYBR Green PCR Master Mix (Applied Biosystems) on the QuantStudio 5 Real-Time PCR Instrument (Applied Biosystems). The mRNA levels of various genes were measured in triplicate and normalized with Gapdh. Information on the oligonucleotides used in this study is provided as Supplementary Table S1.

**Histological Analysis**. For cartilage immunofluorescence staining, the tissues were fixed with 4% paraformaldehyde (Wako) in 0.01 M PBS (pH 7.4) overnight at 4°C, followed by decalcification using 10% EDTA solution. After being embedded in paraffin (Leica Biosystems), the samples were sectioned at a thickness of 6 µm. The tissue sections were incubated with the primary antibodies against Mast4 (Bioworld Technology), Col2a1 (Abcam), and Sox9 (Cell Signaling Technology) at 4°C overnight. After washing in PBS, the tissue sections were consecutively incubated in AlexaFluor488 (Invitrogen) for 2 h at room temperature. Then the tissue sections were counter-stained with TO-PRO™−3 (Invitrogen) for 15 minutes. The images

were taken using a confocal microscope DMi8 (Leica). To detect collagen tissue, sections were stained with freshly prepared Russell-Movat modified pentachrome (American MasterTech) according to the manufacturer's protocols. The images were made binary at a standard threshold, and the positive pixels were counted by using the Leica Microsystem CTR 6000 (Leica). For bone immunofluorescence staining, the mice were anesthetized and perfusion-fixed with 4% PFA to collect femurs and tibiae. The samples were fixed with 2% PFA at 4 °C overnight. The samples were decalcified in 0.5 M EDTA solution for 6 days. Then, the samples were embedded into 5% low melting agarose (Invitrogen) and cut into 150 µm sections by vibratome (Leica, CT1200S). After removal of agarose from the sections, the sections were permeabilized with PBST (0.3% Triton X-100 in phosphate-buffered saline) for 20 minutes and blocked with 5% goat serum in PBST for 30 minutes. The sections were incubated with primary antibodies diluted in blocking solution at RT for 2 h, washed for 3 times with PBS and treated with secondary antibodies in blocking solution at RT for 75 minutes. After the sections were washed in PBST for 3 times and PBS for 3 times, the sections were mounted on microscope glass slides with fluorescence mounting medium (DAKO). Primary antibodies and reagents used for immunofluorescence were as follows: CD31 (Millipore, MAB1398Zm, 1:150), MMP13 (Abcam, 1:150), Osterix (Abcam, 1:300), Runx2 monoclonal (Cell Signaling Technology, 1:150). Secondary antibodies and reagents used for IF were as follows: FITC-conjugated anti-hamster IgG (Jackson ImmunoResearch, 1:300), Cy3-conjugated anti-rabbit IgG (Jackson ImmunoResearch, 1:300). Stained bone sections were analyzed at high resolution with a Zeiss LSM 880 confocal microscope (Carl Zeiss). Z-stacks of images were processed with Zen software.

**Elemental Mapping by EPMA**. An electron probe microanalyzer (EPMA-1610; Shimadzu, Kyoto, Japan) was used for the elemental mapping of Ca, P, and Mg. Undecalcified 6-week-old mouse tibias were embedded in epoxy resin and trimmed with diamond disks until exposure to a sagittal plane. After polishing, the specimens were sputter-coated with carbon before elemental analysis. For each experiment, $256 \times 256$ pixels mapping were performed. The accelerating voltage and beam current were set to 15 kV and 0.03 µA, respectively, and integrating time was 0.05 seconds at each pixel.

**Micro CT**. Three-dimensional reconstructed computed tomography images were obtained by scanning calcified SPC-generated bone regions with a MicroCT, Skyscan 1076 (Antwerp). The data were then digitalized using a frame grabber, and the resulting images were transmitted to a computer for analysis using Comprehensive TeX Archive Network (CTAN) topographic reconstruction software.

**In vivo calcein labeling**. Three-week-old mice were intraperitoneally injected with 50 mg/kg of calcein (Sigma-Aldrich, St. Louis, MO) in a 5% sodium bicarbonate solution. Mice were labeled 7 days and 2 days prior to sacrifice. Tibias were fixed in 4% paraformaldehyde for 1 day at RT. Samples were incubated in 10% (v/v) KOH for 96 h and embedded in paraffin, as previously described[63]. Embedded samples were sectioned in 5 µm thickness and visualized with confocal microscope (DMi8, Leica, Germany). Distance between the labels on cortical bone was measured at 3 points per sample.

**Growth plate morphometry**. For morphometric analysis, the total thickness of the growth plate cartilage at the proximal end of each tibia was measured at the H&E- or pentachrome-stained section images, equally spaced intervals along an axis oriented 90° to the transverse plane of the growth plate and parallel to the longitudinal axis of the growth plate. Three measurements were obtained from each epiphyseal growth plate, and final thickness determinations in individual animal indicated the average of these values using image-analysis software (ImageJ, ver. 1.38e, NIH, USA). The widths of the layers occupied by hypertrophic chondrocytes were measured by the same method. In addition, the percentage of the hypertrophic layer to the total thickness of the growth plate was calculated. Three left and right tibias were used for each group.

**Isolation of mouse skeletal stem cells using flow cytometry**. FACS separation was performed, referring to the protocol[27]. Briefly, male Mast4$^{+/+}$ and Mast4$^{-/-}$ mice (n = 5) at 5 weeks of age were sacrificed, followed by dissection of humerus, femur and tibia. Cells were isolated with a combination of mechanical and chemical digestion, and red blood cells were removed by ammonium–chloride–potassium (ACK) lysis buffer. The TER119$^+$CD45$^+$ hematopoietic cells were filtered by magnetic-activated cell sorting (MACS). The remaining cells were then stained for the following antibodies: CD45, TER119, TIE2, ITGAV, CD202B, THY1.1, THY1.2, CD105, and 6C3. 7-AAD was used for live/dead cell discrimination. FACS analysis was performed on an FACS Aria II Instrument (BD Biosciences) and analyzed by FlowJo v10.7.1 and BD FACSDiva v9.0.1 software.

In vivo **cartilage formation assay**
The control and Mast4-depleted C3H10T1/2 cells were cultured in the chondrogenic differentiation medium, including BMP-2 (150 ng/ml), for 4 days in a micromass culture. The cells were resuspended in PBS (100 micromass cultures in 100 µl per injection) and subcutaneously injected into the flanks of athymic nude

mice (6-week old females; $n = 4$). After 2 weeks, the mice were euthanized, and the grafts were collected for IHC evaluation. The volume of cartilage-containing grafts was measured and calculated using the formula $V = (A*B^2)/2$, where $V$ is volume (mm$^3$), $A$ is long diameter (mm), and $B$ is short diameter (mm). All experiments were conducted in accordance with guidelines provided by the Institutional Animal Care and Use Committee of Center for Phenogenomics Animal Research Facility, Woojung BSC (Suwon, Korea, Association for Assessment and Accreditation of Laboratory Animal Care-accredited facility).

**In vivo transplantation in full-thickness cartilage defect rabbit model.** A full-thickness cartilage defect model was prepared as previously reported[64]. Briefly, thirteen healthy New Zealand white male rabbits (3.0-3.5 kg in weight) were obtained 4 weeks before the experiment. The rabbits were anesthetized with Zoletil and xylazine. In sterile conditions, a parapatellar skin incision skin incision was made on the right knees, and the patella was dislocated laterally. Full-thickness osteochondral defects (3 mm in diameter and 3 mm in depth) were created at the center of the trochlear groove of the femur by drilling. Cartilage and bone debris were removed, and the defect sites were carefully washed with normal saline. Vehicle (PBS 50 μl), naïve or MAST4-depleted hBMSCs ($2 \times 10^6$ cells in 50 μl; passage 6-7) were transplanted into the defect sites ($n = 3$ for vehicle, $n = 5$ for naïve and MAST4-depleted hBMSCs), followed by relocation of the patella. The wound was closed with 4-0 nylon sutures. All procedures were conducted in accordance with guidelines provided by the Institutional Animal Care and Use Committee of KNOTUS Co., Ltd (Incheon, Korea).

**Chromatin immunoprecipitation.** Cells were cross-linked with 1% formaldehyde for 10 minutes at room temperature. Glycine was added to a final concentration of 125 mM for 5 minutes to quench the formaldehyde crosslinks. Cells were washed with ice-cold phosphate buffered saline, harvested by scraping, pelleted, and resuspended in SDS lysis buffer (50 mM Tris-HCl [pH 8.1], 1% SDS, 10 mM EDTA) with complete protease inhibitor cocktail (Roche). Cell extracts were sonicated with a Bioruptor TOS-UCW-310-EX (output, 250 W; 23 cycles of sonication with 30-second intervals; Cosmo Bio). Samples were centrifuged at $18,472 \times g$ at 4°C for 10 minutes, and the supernatants were diluted 10-fold in dilution buffer (20 mM Tris-HCl [pH 8.0], 2 mM EDTA, 1% Triton X-100, 150 mM NaCl, and complete protease inhibitor cocktail). Chromatin samples were precleared with protein A-agarose beads (Santa Cruz) for 2 h before immuno-precipitation against Sox9 and Smad3 (Abcam) antibodies overnight at 4°C. Immune complexes were collected with protein A-agarose beads. Samples were washed five times (first wash with low salt immune complex wash buffer [20 mM Tris-HCl, pH.8.0, 2 mM EDTA, 1% Triton X-100, 0.1 % SDS, and 150 mM NaCl], second wash with high salt immune complex wash buffer [20 mM Tris-HCl, pH.8.0, 2 mM EDTA, 1% Triton X-100, 0.1% SDS, and 500 mM NaCl], third wash with LiCl immune complex wash buffer [10 mM Tris-HCl, pH.8.0, 1 mM EDTA, 250 mM LiCl, 1% NP-40, and 1% Na-deoxycholate], and the last two washes with TE buffer). Immunoprecipitated samples were eluted with buffer containing 1% SDS and 100 mM NaHCO₃ at room temperature. Eluates were heated overnight at 65°C to reverse crosslinks after adding NaCl to a final concentration of 100 mM. Genomic DNA was extracted with a PCR purification kit (GeneAll). Precipitated chromatin by real-time PCR and the readouts were normalized using 5% input chromatin for each sample. The experiments were repeated two or more times. A forward primer of 5'-AACCCTGCCCGTATTTATTT-3' and a reverse primer of 5'-TGTGCATTGTGGGAGAGG-3' were used to detect the binding of Sox 9 to the *Col2a1* gene. A forward of 5'-TGCTGACACTTTATTTTGCTCT-3' and a reverse primer of 5'-CATCTCCAAGCCTCTTTCTG-3' were used to detect the binding of Smad3 to the *Mast4* gene.

**Ubiquitination assay.** Flag-MAST4-PDZ, GFP-Smruf1, GFP-GSK-3β, and HA-Ub plasmids were transfected into C3H10T1/2 cells, followed by MG-132 treatment (10 μM for 6 h). Cells were lysed in SDS lysis buffer [10 mM Tris-HCl (pH 8.0), 150 mM NaCl, 1% SDS, 5 mM NEM, protease inhibitor] by boiling for 10 min, followed by 10-fold dilution with dilution buffer [10 mM Tris-HCl (pH 8.0), 150 mM NaCl, 1% Triton X-100]. Lysed samples were immunoprecipitated with Flag antibody (Sigma-aldrich) overnight, and antibody-bound proteins were precipitated with Dynabeads. Washing buffer A [10 mM Tris-HCl (pH 8.0), 150 mM NaCl, 1% Triton X-100, 0.1% SDS] and B [10 mM Tris-HCl (pH 8.0), 150 mM NaCl, 1% Triton X-100] were used to wash precipitated samples, followed by western blotting.

**Luciferase assay.** C3H10T1/2 cells were transiently transfected with 4xCol2a1-luc, Smad3/4-responsive promoter (CAGA)₁₂-luc, SBE-luc, 6xOSE-luc, MAST4-promoter luciferase report plasmids, HA-MAST4 PDZ, Myc-Sox9 WT/S494A/S494D plasmids using polyethylenimine (Polysciences). Cells were treated with TGF-β1 (3 ng/ml for 24 h) (R&D Systems) and Vactosertib (500 nM for 26 h). The luciferase activities were analyzed using the Luciferase Assay System kit (Promega) according to the manufacturer's protocol. All assays were done in triplicate, and all values were normalized for transfection efficiency against β-galactosidase activities.

**Immunoprecipitation assay and western blot analysis.** For the immunoprecipitation assay, cell extracts were incubated with the indicated primary antibodies overnight at 4°C. Antibody-bound proteins were precipitated with Dynabeads Protein G (Invitrogen). Cells were lysed in a RIPA buffer containing protease inhibitor cocktail (Complete; Roche). Samples were separated by SDS-PAGE, followed by electrotransfer to polyvinylidene difluoride membranes (PVDF; Millipore). The membrane was blocked for 1 h at room temperature and incubated overnight at 4°C with the primary antibodies. Horseradish peroxidase-conjugated antibodies (Millipore) were used as secondary antibodies. The peroxidase reaction products were visualized with WESTZOL (Intron). All signals were detected by Amersham Imager 600 (GE Healthcare Life Sciences). Information on the antibodies used in this study is provided as Supplementary Table S2.

**In-gel digestion for mass spectrometry analysis sample preparation.** The gel band corresponding to Myc-Sox9 size was excised and destained for 15 min with 50% (v/v) acetonitrile (ACN) prepared in 25 mM ammonium bicarbonate, and 100 mM ammonium bicarbonate sequentially. Proteins were reduced with 20 mM DTT at 60 °C for 1 h and then alkylated with 55 mM iodoacetamide at room temperature for 45 min in the dark. After dehydration, the proteins were digested with Trypsin/Lys-C Mix, mass spec grade (Promega, Madison, WI, USA) prepared in 50 mM ammonium bicarbonate overnight at 37 °C. The peptides were extracted from the gel pieces with 50% (v/v) ACN prepared in 5% formic acid, dried under a Centrivap concentrator (Labconco, Kansas City, MO, USA), and stored at −20 °C until use.

**Mass spectrometry for the detection of phosphorylation of Sox9.** The peptide samples extracted by in-gel digestion were suspended in 20 μl of solvent A (0.1% formic acid prepared in water, Optima LC/MS grade, ThermoFisher Scientific). Thereafter, 4 μl of the sample was loaded onto a EASYSpray C18 column (75 μm × 50 cm, 2 μm) and separated with a 2–35% gradient of solvent B (0.1% formic acid prepared in ACN) for 65 min at a flow rate of 300 nL/min. Mass spectra were recorded on a Q Exactive hybrid quadrupole-Orbitrap mass spectrometer (Thermo Fisher Scientific) interfaced with a nano-ultraHPLC system (Easy-nLC1000; Thermo Scientific). The spray voltage was set to 1.5 kV and the temperature of the heated capillary was set to 250 °C. The Q-Exactive was operated in data-dependent mode and each cycle of survey consisted of full MS scan at the mass range 300-1400 m/z and MS/MS scan for ten most intense ions. Exclusion time of previously fragmented peptides was for 20 sec. Peptides were fragmented using Higher energy collision dissociation and the normalized collision energy value was set at 27%. The resolutions of full MS scans and MS/MS scans were 70,000 and 17,500. The advanced gain control target was $5 \times 10^4$, maximum injection time was 120 ms, and the isolation window was set to 3 $m/z$.

The raw data were processed by using the Trans-Proteomic Pipeline (v4.8.0 PHILAE) for converting to mzXML file which is search-available format. Database search for sequenced peptides was the Sequest (version 27) algorithm in the SORCERER (Sage-N Research, Milpitas) platform with Uniprot human database. Parent and fragment ion tolerance were set to 10ppm (monoisotopic) and 1 Da (monoisotopic), respectively. Fixed modification was set on cysteine of 57 Da (carbamidomethylation). Variable modifications were set on methionine of 16 Da(oxidation) and on serine, threonine, tyrosine of 80 Da (phosphorylation). Trypsin was chosen as an enzyme with a maximum allowance of up to two missed cleavages. The Scaffold software package (version 3.4.9, Proteome Software Inc., Portland, OR, USA) was used to validate MS/MS-based peptide and protein identifications. The thresholds for peptide and protein identification were 95% minimum and 95% minimum, 2 peptides minimum, respectively. Peptide and protein FDR were 0.2% (Decoy) and 0.6% (Decoy).

**Statistical analyses.** All quantitative experiments were performed in triplicate and/or repeated at least three times. Data were expressed as mean ± SD. Student t tests was conducted using GraphPad Prism version 5 (GraphPad Software Inc.). $P < 0.05$ was considered statistically significant. Significance was achieved at $P < 0.05$.

**Reporting summary.** Further information on research design is available in the Nature Research Reporting Summary linked to this article.

## Data availability

The RNA sequencing data generated in this study have been deposited in the NCBI SRA database under accession code SRR12095157 and SRR12095158 for Figs. 1c, f, Supplementary Figs. 5 and 6, SRR12095360 and SRR12095361 for Figs. 6b, c, Supplementary Figs. 25 and 26c, SRR12095362 and SRR12095363 for Figs. 6a, c, Supplementary Figs. 25 and 26a. Source data are provided with this paper. The source data for Figs. 1b, d, e, 2c, d, j, 3b, d, e, 4l, 5e, f, 7a, d and Supplementary Figs. 2d, 3b, 7b, 8b, 9a, b, 10, 11b, c, 12a, 13b, 17, 19c, d, e, 20b, d, 23b, c, 26b, d, 28b, c, and 29b have been provided as Source Data file. Unprocessed original scans of blots in Figs. 1a, b, 2b, e, f, g, i, l, 3a, c, f, g, 4b, c, d, f, g, h, i, j, k, 6d, e are shown in Source Data file. Unprocessed original scans of blots in Supplementary Figs. 2b, c, e, 4b, 7e, 9c, d, e, 12b, 14, 15c, d, e, 16b, c, 18d, and 24c and the mass spectrometry data in Fig. 2h are shown at the end of the Supplementary Information file. Source data are provided with this paper.

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

## Acknowledgements
We thank J. Letterio and B. Park for critical review of the manuscript, H.B. Kim and E.C. Yi for MASS SPEC analysis. HA-Sox9 and 4xCol2a1-luc plasmid were kindly provided by R. Nishimura (Osaka University, Japan). This research was supported by a grant of the Basic Research Support Project funded by GILO Foundation, Republic of Korea. The RNA sequencing and data analyses have been supported in part by Theragen Etex Co. Ltd. In vivo transplantation analysis using a full-thickness cartilage defect rabbit model was supported by Amoris Bio Inc. MASS SPEC analysis was supported in part by the National Research Foundation of Korea (NRF) Grant funded by the Korean Government (MSIP) (NRF-2016R1A5A1010764 and NRF-2015M3A9B6073840 to J.W.C). The research was supported in part by the National Research Foundation of Korea (NRF) Grant funded by the Korean Government (MSIP) (NRF-2022R1A2B5B03001627 and NRF-2016R1A5A2008630 to H.-S. J.) and the Bio & Medical Technology Development Program of the National Research Foundation (NRF) & funded by the Korean Government (MSIP&MOHW) (No. 2017M3A9E4048172 to H.-S. J.).

## Author contributions
S.J.K. conceived and supervised the project and wrote the manuscript. S.T. and H.J. designed the study, supervised the project, and made substantial contributions to the final manuscript. P.K., J.P., D.L., and S.M. designed experiments, collected the data, performed the analysis, and wrote the manuscript. C.H. conducted RNA sequencing and bioinformatics analyses. M.S., M.T., Y.J., R.Y., H.P. and H.O. contributed to data collection. K.Y., S.P., and K.P. contributed to data analysis and interpretation. M.L., S.J., H.K., and S.L. performed histological analyses. S.S. conducted in vivo transplantation in full-thickness cartilage defect rabbit model. J.C. contributed to acquisition of MASS SPEC analysis. S.M., F.S., and S.T. generated the MAST4 knockout mice. All authors discussed the results and gave final approval of the version of the manuscript to be submitted.

## Competing interests
The authors declare the following competing interests: S.J.K. has a personal financial interest as a shareholder in Amoris Bio Inc. and TheragenEtex; and declares ownership in Medpacto Inc. J.P. is an employee of Amoris Bio Inc., and C.H. is an employee of TheragenEtex, and K.Y. is an employee of Medpacto Inc. S.S. declares ownership in SCM Lifescience Inc. S.T, S.J.K., and H.J are listed as inventors on the pending patent application(s) on the modulation of MSCs by Mast4 deletion. All other authors declare no competing interests.
