## [Peer Review File · Nature Communications]

Mast4 determines the cell fate of MSCs for bone and cartilage developmentREVIEWER COMMENTS

Reviewer #1 (Remarks to the Author):

In the manuscript "Suppression of Mast4 gene transcription by TGF- β 1 induces chondrogenesis of MSCs and increase of Mast4 stability by Wnt induces osteogenesis of MSCs", demonstrated that the microtubule-associated serine/threonine kinase 4 (Mast4) is a key regulator of chondro-osteogenic differentiation of MSCs. Serine 494 phosphorylation of Sox9 by Mast4 induces proteasomal degradation of Sox9, suppressing chondrogenesis while TGF- β 1-induced suppression of Mast4 expression increases Sox9 stability and Sox9-Smad3 association, resulting in chondrogenesis. Alternatively, GSK3 β -induced phosphorylation of Mast4 recruits the ubiquitin E3 ligase Smurf1 and induces proteasomal degradation of Mast4. Wnt-induced suppression of GSK3 β stabilizes Mast4 and increases β -catenin level and Runx2 activity, resulting in osteogenesis. These hypotheses were clearly and well supported by thorough biochemical analyses, two independent unbiased transcriptome analyses, and in vivo studies using mice with CRISPR/Cas9-mediated germline deletion of Mast4. Finally, using a rabbit full-thickness cartilage defect model, therapeutic effects of Mast4-deficient human BMSCs on cartilage repair were provided. novel functions of Mast4 in the commitment of MSC to chondro-osteogenic differentiation, the soundness and thoroughness of the experimental approach, and potential MSC-based therapy for cartilage regeneration are very appealing and appropriate for publication in the Nature Communications.

Major points:

1. Functional analyses of Mast4 were mainly performed in the MSC-like cell line C3H10T1/2 using CRISPR/Cas9-mediated deletion of Mast4. The author should isolate skeletal stem cells from Mast4 (+/+) and Mast4 (-/-) pups using the expression of cell surface markers (PMID: 29748647) and perform in vitro TGF β 1-induced chondrogenesis and Wnt-induced osteogenesis of these cells to confirm the data seen in Mast4-deficient C3H10T1/2 cells.
2. In Fig. 4, the author should demonstrate that ubiquitination of HA-Mast4-PDZ is upregulated in the presence of Smurf1 and/or GSK-3 β and that this ubiquitination disappears in HA-Mast4-PDZ-d632-636.

Minor point:

1. Fig. 5f needs mouse sample numbers and standard deviation.

Reviewer #2 (Remarks to the Author):

In this manuscript, a novel role for the kinase Mast4 is reported upstream of chondrogenic and osteogenic pathways. Through an extensive series of well-performed in vitro and in vivo experiments, the authors propose a model whereby this kinase regulates Sox9 stability via S494 phosphorylation. Moreover, Mast4 stability itself is regulated by Smurf1-mediated ubiquitination, a process inhibited by GSK3 β downstream of WNT signaling. A novel Mast4 $^{-/-}$ strain is reported, and the authors perform 'translational' studies looking at Mast4 depletion in a rabbit cartilage injury model. Overall, enthusiasm for these novel and important studies is high- this work clearly identifies Mast4 as a factor that regulates chondrogenesis and presents a novel mechanism underlying this function.

Major points:

1. Very little insight is provided as to why the authors started studying Mast4 in the first place. Was this identified by RNAseq on developing chondrocytes? More transparency is needed regarding the origins of this work.
2. Throughout the manuscript, the authors use the somewhat controversial term "MSC". In

general, most investigators use this term to note bone marrow stromal cells cultured ex vivo under specific conditions. However, the physiologic relevance of "MSCs" to normal bone/cartilage development and repair is not known. For this reason, the term skeletal stem cell/progenitor is preferred (see PMID 31572721).

3. The model proposed in extended data figure (EDF) 29 is clear and appealing based on this work. However, the manuscript's abstract and introduction would benefit from considerable revision to better set the stage for the model/conclusions that follow in this manuscript. For example, the authors should clearly indicate that Mast4 is a kinase in the abstract. Furthermore, the introduction could better highlight the mechanistic data to follow regarding GSK3b and Sox9.

4. The overall phenotype of the novel Mast4^{-/-} mice should be better described. Do these mice have shorter limbs? Is Mast4 expressed outside of bone/cartilage? If so, could there be other reasons to explain the small overall size of the mutant mice?

5. Along the lines of point #4, quantification of the bone mass results shown in Figure 5f is needed. These graphs have no error bars or statistical significance. Also- analysis of cortical bone is needed to better understand the changes seen in trabecular bone.

6. The final experiment using human cells transplanted into rabbits is quite important and interesting. The authors should move the quantification of these results into the main text figures. Each animal should be shown as an independent data point (rather than the box/plunger plots shown).

7. Along these lines, have the authors measured articular chondrocyte thickness in the Mast4^{-/-} mice? It seems that their current analyses are restricted to growth plate chondrocyte measurements.

Minor points:

1. For EDF3, quantification of the Sox9 mRNA and protein levels are needed to support the claim that Mast4 preferentially regulates the protein

2. For EDF9a, it would be ideal to show Sox9 ChIP in Mast4 overexpressing cells. For these ChIP studies, have the authors considered effects of Mast4 on total cellular Sox9 levels?

3. For EDF9c and EDF9d, quantification of immunoblotting results is needed, especially since MG132 seems to increase Sox9 protein levels even in cells without Mast4 over-expression

4. For Figure 2E, the authors cannot claim direct protein/protein interaction based on cell-based co-immunoprecipitation results. To do this, recombinant proteins are needed.

5. Figure 2G does not include MG132 treatment even though the text indicates that it does

6. For EDF10A, the changes in Sox9 S181 and S199 phosphorylation are quite modest. Why do the authors think that these changes are 'real'? Were replicates performed?

7. For Figure 2I- it is not totally clear why the S494D phospho-mimetic form of Sox9 still responds to Mast4 over-expression. The authors should provide a potential explanation for this finding.

8. For EDF21B, the authors should provide quantitative measurement of their back-scatter electron imaging results. Alternatively, the authors could simply perform cortical bone microCT results to measure cortical thickness and cortical bone tissue mineral density.

Reviewer #3 (Remarks to the Author):

Comments to the Author

The authors are writing to show that (1) Mast4 is suppressed by TGF-beta1, (2) Mast4 RNA is stabilized by Wnt, and (3) Mast4 enhanced Sox9 degradation and transition of chondrocytes to osteoblasts. Mast4 was identified as significantly suppressed by Tgf-beta1 which led to the study for osteo-chondrogenic effects described here.

This is an interesting and novel development and would provide another tool for increasing cartilage synthesis in vivo, "simply apply Mast4 depleted MSCs" and the authors make an initial use of this for their paper, making the work not only novel but potentially significant. In the literature, Mast4 is known to be involved in Fgf2 signaling, per Lee et al., Cell Death Diff, 2021, and is a target for O-GlcNAc modification in osteoblasts (Nagel et al., Mol Cell Proteomics, 2013). Furthermore, Mast4 is known to inhibit Foxo1 in brain where it's best characterized (Gongol et al., Sci Rep, 2017). The role of Mast4 in osteoblast/chondrocyte lineage determination is the key to

the novelty of this paper and is what makes the work so exciting.

The methods used for this paper are standard *in vitro* and *in vivo* (mouse) techniques, including some deep sequencing (RNA-seq), viral gain of function studies, generation of novel Mast4 knockout mice, as well as staining of samples from the mice under investigation. The methods are robust and appropriate to the studies being performed.

The figures can be described as follows. Figure 1 shows that Mast4 is consistently downregulated during chondrogenesis, as shown using multiple methods, and that Mast4 knockout functionally inhibits osteoblastogenesis and enhances chondrogenesis related gene expression *in vitro*. Figure 2 shows Mast4's interaction with Sox9 post-translationally to permit its degradation using ChIP-qPCR, luciferase, and *in vitro* methods with chondrogenesis. It is important to show that some of these things can be done with Mast4 knockdown *in vitro*, especially the *in vitro* effects, using shRNA rather than purely the genomic knockout. Figure 3 shows Mast4 suppression by Tgf-beta1 and enhanced chondrogenesis through increased Sox9-Smad3 association, using HA pull-down and co-IP methods. Figure 4 shows Wnt induction of osteogenesis by enhancing Mast4 stability through inhibition of Gsk-3beta, using various specific mutants of Mast4 where mutants inhibit the effects observed, also, inhibition of Gsk-3beta enhances Mast4. That this change in Mast4 activity changes beta-catenin activity is confirmed with luciferase assay. Figure 5 shows *in vivo* results of Mast4 knockout with increased cartilage and osteoporosis. The figure currently includes micro-CT, and considerable immunofluorescence imaging/staining, however, microCT with hexabrix would be important for demonstrating changes in cartilage with the knockout of Mast4. Figure 6 shows an analysis of differentially expressed genes with Mast4 knockout from mouse bone and cartilage. It also uses RT-PCR, but could be enhanced with quantification using qPCR (e.g. SYBRgreen) rather than RT-PCR and gel imaging. Figure 7 shows some clinical utility (or at least adjacent) use where Mast4 knockout MSCs can be used to repair cartilage *in vivo* with a cartilage defect model and the injection of transgenic MSCs from an MSC cell line (C3H10T1/2).

Overall, the paper is a solid paper, and mostly well written, with good data on an interesting topic. However, the manuscript needs to do a Major Revision to address the major concerns and minor concerns listed below before it can be published in Nature Communications.

Major concerns:

- The introduction needs some introduction to Mast4 and its prior description in the literature. It is known to have a role in a few osteoblast relevant signaling pathways (e.g. FGF2, per Lee et al., Cell Death Diff, 2021).
- Related to the prior point, it is important to look at and address previous roles that may potentially be affecting the currently described role.
- Figure 2 needs shRNA knockdown of Mast4 to confirm that the effects of genomic knockout aren't unique to the particular knockout used.
 - o This shRNA knockdown needs to be included for all key experimental findings where other knockdown/knockout methods are used – especially the experiments showing that Mast4 KO results in increased cartilage formation (Figure 2) and decreased osteoblastogenesis (Figure 4).
- Figure 5 needs microCT confirmation of changes in cartilage and hexabrix analysis is particularly important for demonstrating changes in cartilage with the knockout of Mast4.
- Figure 5 also needs confirmation of changes in bone formation using time separated dual staining with Calcein (and/or Alizarin red) injection to show changes in bone growth between WT and Mast4 knockout mice *in vivo*, e.g. Rauch et al. (PLoS One, 2010) or Su et al. (JCI, 2012).
- Figure 6 needs qPCR (e.g. SYBRgreen) with quantification of multiple replicates rather than RT-PCR for this critical analysis and confirmation.
- The truncated overexpression of Mast4 should have more justification (beyond "Due to high molecular weight (>285 kDa) and relatively low expression of full-length Mast4, we alternatively used a truncated Mast4 construct (Mast4-PDZ)" and extended data figure 4) and/or shRNA-knockdown should be used to know that effects are dependent on Mast4 dose.
- Western blot should be used to detect the protein level changes and signaling pathways changes in mutant mice compared with Wild-Type mice, including genes associated with cartilage development in the cartilage of Mast4^{-/-} mice and genes associated with skeletal system development and Wnt signaling pathway in the bone of Mast4^{-/-} mice compared that in Mast4^{+/+}.

Minor concerns:

- The abstract seems to have obviously been written too quickly and should include at least a

sentence providing background by answering the question, "Why are we looking at Mast4?"

- Figure legends shouldn't include results, simply descriptions of what is being done. Descriptions of results should be limited to the results and figure legend titles.
- Primer sequences need to be provided for each RT-PCR and/or any qPCR performed.
- Antibody catalog numbers and/or clone IDs need to be provided for the antibodies used in the western blots performed.

Reviewer #1:

Major concerns:

1. Functional analyses of *Mast4* were mainly performed in the MSC-like cell line C3H10T1/2 using CRISPR/Cas9-mediated deletion of *Mast4*. The author should isolate skeletal stem cells from *Mast4* (+/+) and *Mast4* (-/-) pups using the expression of cell surface markers (PMID: 29748647) and perform *in vitro* TGF β 1-induced chondrogenesis and Wnt-induced osteogenesis of these cells to confirm the data seen in *Mast4*-deficient C3H10T1/2 cells.

We appreciate Reviewer #1's critical comment concerning isolation and functional assessment of skeletal stem cells from *Mast4*^{+/+} and *Mast4*^{-/-} mice. As Reviewer #1 suggested, we isolated mouse skeletal stem cells, a purified population of CD45⁺TER119⁻TIE2⁻ITGAV⁺THY1⁺6C3⁻CD105⁻, from 5-week-old *Mast4*^{+/+} and *Mast4*^{-/-} mice (n=5) following the flow cytometry-based protocol provided by Reviewer #1. After culturing the isolated cells under hypoxic conditions for 3 weeks to form the colonies and staining the colonies with either alcian blue or Alizarin Red S, we confirmed enhanced chondrogenic but reduced osteogenic differentiation of the skeletal stem cells isolated from *Mast4*^{-/-} mice (See Fig. A). These results indicate that *Mast4* regulation of chondrogenic or osteogenic differentiation can also be applied to skeletal stem cells as shown in MSC-like cell line C3H10T1/2 and bone marrow-derived mesenchymal stem cells. Although we tried to provide mRNA expression of lineage-specific markers in the colonies, it was difficult to obtain a sufficient amount of RNA from the colonies formed. Instead, we extracted the stained colonies for absorbance measurements and included graphs showing the absorbance values. In addition to the results obtained from the bone marrow-derived mesenchymal stem cells isolated from *Mast4*^{-/-} mice (EDF 24), we believe the results obtained from skeletal stem cells will strengthen our study on the role of *Mast4* on chondro/osteogenesis of stem cells. We appreciate Reviewer #1's comment that led to the improvement of our data. **We added these results to Fig. 5h and EDF22-23 and mentioned in the manuscript (p.10).**

Fig. A. Isolation and functional assessment of mouse skeletal stem cells from 5-week-old *Mast4*^{+/+} and *Mast4*^{-/-} mice. (a) Representative FACS plots. (b) Alcian blue and Alizarin Red S staining results obtained from *in vitro* colony formation assay using the isolated skeletal stem cells.

2. In Fig. 4, the author should demonstrate that ubiquitination of HA-*Mast4*-PDZ is upregulated in the presence of *Smurf1* and/or *GSK-3 β* and that this ubiquitination disappears in HA-*Mast4*-PDZ-d632-636.

As Reviewer #1 pointed out, we conducted ubiquitination assay in C3H10T1/2 cells co-transfected with HA-Ub and either Flag-Mast4-PDZ wild-type (WT) or $\Delta 632-636$ plasmids in the presence of GFP-Smurf1 and/or GFP-GSK-3 β . In consistent with our previous results showing Smurf1 and GSK-3 β regulation of Mast4 expression, ubiquitination of Mast4-PDZ WT was increased by Smurf1 and further enhanced in the presence of GSK-3 β , while that of Mast4-PDZ $\Delta 632-636$ was affected by neither Smurf1 nor GSK-3 β (see **Fig. B**). This result supports our hypothesis that GSK-3 β -mediated phosphorylation triggers Smurf1 recruitment to Mast4, resulting in ubiquitination and subsequent proteasomal degradation of Mast4. **This result was added to Fig. 4j and mentioned in the manuscript (p.9).**

Fig. B. Smurf1-mediated ubiquitination of Mast4 in conjunction with GSK-3 β in C3H10T1/2 cells.

Minor concerns:

1. *Fig. 5f needs mouse sample numbers and standard deviation.*

As Reviewer #1 suggested, we measured trabecular bone volume and bone mineral density (BMD) with increased number of samples. The tibias of 6-week-old male mice (n=8 for *Mast4*^{+/+} and 10 for *Mast4*^{-/-} mice) were measured. Consistently, both the trabecular bone volume and BMD in the tibias of *Mast4*^{-/-} mice were significantly lower than those of *Mast4*^{+/+} mice (see **Fig. C**). Incidentally, we realized the BMD unit was mistakenly as labeled as mg/cm³, so we corrected it to g/cm³. **We removed the original graphs from Fig. 5f and added new ones to Fig. 5e.**

Fig. C. The trabecular bone volume and bone mineral density in the tibias of *Mast4*^{+/+} and *Mast4*^{-/-} mice at 6 weeks of age.

Reviewer #2:

Major concerns:

1. *Very little insight is provided as to why the authors started studying Mast4 in the first place. Was this identified by RNAseq on developing chondrocytes? More transparency is needed regarding the origins of this work.*

We appreciate Reviewer #2's comment on the provision of the origins of the present study on Mast4. TGF- β has been reported as a key initiator and mediator of chondro-osteogenesis of mesenchymal progenitor cells throughout differentiation stages, and we sought to identify a novel gene that may be involved in the regulation of switching mesenchymal progenitor cells to specific lineages downstream of TGF- β signal. Considering that TGF- β signal is activated by serine/threonine receptor kinases and protein kinases play a crucial role in signal transduction, we focused on genes encoding protein kinases that also showed considerable expression level in the RNA sequencing of C3H10T1/2 cells. Through RT-PCR screening of nearly 30 genes of whether their expression was regulated by TGF- β , we found that Mast4 was significantly down-regulated by TGF- β in C3H10T1/2 cells. Moreover, all the MAST family members, except Mast1, showed considerable expression level. While developing our study on Mast4 based on these findings, we observed that Mast4 was regulated by not only TGF- β , but also Wnt and mediated chondrogenic or osteogenic differentiation of C3H10T1/2 cells. In regards to Reviewer #2's suggestion, **we have provided an explanation in the manuscript (p.3)**

2. *Throughout the manuscript, the authors use the somewhat controversial term "MSC". In general, most investigators use this term to note bone marrow stromal cells cultured ex vivo under specific conditions. However, the physiologic relevance of "MSCs" to normal bone/cartilage development and repair is not known. For this reason, the term skeletal stem cell/progenitor is preferred (see PMID 31572721).*

We appreciate Reviewer #2's comment on the use of term "MSC." As Reviewer #2 is rightfully concerned, we have also agonized over the use of this controversial term "MSC" because we acknowledge that currently-termed MSCs are a heterogenous population containing only a fraction of cells with self-renewal potential and multipotency, thereby being preferred to be called 'mesenchymal stromal cells.' However, considering that we have mainly used C3H10T1/2 mouse embryo fibroblasts, often regarded as mouse mesenchymal stem cell line, and mouse/human bone marrow-derived stromal/stem cells in our experiments and since chondrogenesis/osteogenesis is often introduced as being derived from mesenchymal stem cell differentiation, we decided to use the term MSC. We hope Reviewer #2 would understand our situation of not being able to replace the term "MSC" with "skeletal stem cells."

However, in order to demonstrate whether the effect of Mast4 on chondrogenic and osteogenic differentiation of previously mentioned MSC is applicable to skeletal stem cells, we isolated mouse skeletal stem cells, a purified population of CD45⁻TER119⁻TIE2⁻ITGAV⁺THY1⁻6C3⁻CD105⁻, from 5-week-old *Mast4*^{+/+} and *Mast4*^{-/-} mice (n=5) following the flow cytometry-based protocol¹. After culturing the isolated cells under hypoxic conditions for 3 weeks, allowing colonies to form and stained them with either alcian blue or Alizarin Red S, we confirmed enhanced chondrogenic but reduced osteogenic differentiation of the skeletal stem cells isolated from *Mast4*^{-/-} mice (see **Fig. A**). These results indicate that Mast4 regulation of chondrogenic or osteogenic differentiation shown in MSC-like cell line C3H10T1/2 and bone marrow-derived mesenchymal stromal/stem cells can also be applied to skeletal stem cells. **These results were added to Fig. 5h and EDF22-23 and mentioned in the manuscript (p.10), using the term "skeletal stem cells."**

¹Gulati, G.S. *et al.* Isolation and functional assessment of mouse skeletal stem cell lineage. *Nat. Protoc* 13, 1294-1309, doi:10.1038/nprot.2018.041 (2018).

Fig. A. Isolation and functional assessment of mouse skeletal stem cells from 5-week-old Mast4^{+/+} and Mast4^{-/-} mice. (a) Representative FACS plots. (b) Alcian blue and Alizarin Red S staining results obtained from *in vitro* colony formation assay using the isolated skeletal stem cells.

3. The model proposed in extended data figure (EDF) 29 is clear and appealing based on this work. However, the manuscript's abstract and introduction would benefit from considerable revision to better set the stage for the model/conclusions that follow in this manuscript. For example, the authors should clearly indicate that Mast4 is a kinase in the abstract. Furthermore, the introduction could better highlight the mechanistic data to follow regarding GSK3b and Sox9.

As Reviewer #2 suggested, we elaborated the abbreviated term Mast4 (microtubule-associated serine/threonine kinase family member 4) and briefly mentioned the importance of studying protein kinases in the abstract. Furthermore, we supplemented the introduction by explaining the mechanisms of GSK-3β-Smurf1-Mast4 and Mast4-Sox9 in more detail (p.3-4).

4. The overall phenotype of the novel Mast4^{-/-} mice should be better described. Do these mice have shorter limbs? Is Mast4 expressed outside of bone/cartilage? If so, could there be other reasons to explain the small overall size of the mutant mice?

In order to better explain the small overall size of Mast4^{-/-} mice, we measured the length of the limbs, including humerus, radius, femur, and tibia, from μCT images of 6-week-old male mice as suggested by Reviewer #2. Consistent with the small overall size of Mast4^{-/-} mice, all the limbs of Mast4^{-/-} mice were significantly shorter than those of Mast4^{+/+} mice (see Fig. B). Furthermore, as shown in Fig. 5f, we found that bone formation, measured by double calcein labelling, in the tibias of Mast4^{-/-} mice at 3 weeks of age was significantly decreased (see Fig. C). Considering these results, reduced bone formation and an altered switch to bone growth in the terminal stage of chondrocyte differentiation mediated by Mast4 depletion may contribute to smaller size of Mast4^{-/-} mice. **These results are included in Fig. 5f and EDF20d and mentioned in the manuscript (p.10).** Moreover, in order to address Reviewer #2's question, we examined the mRNA and protein expression of Mast4 in different murine organs besides bone and cartilage. We observed a high expression of Mast4 in other organs such as brain, kidney and lung (see Fig. D). We acknowledge the possibility that Mast4 effects on these organs may also contribute to the small overall size. Thus, we plan to investigate the role of Mast4 in these organs in the near future. Therefore, we have decided to exclude this results in the present study,

although we briefly mention it in the manuscript (p.9). We hope Reviewer #2 understand our intention to highlight the role of Mast4 on MSC differentiation into cartilage/bone.

Fig. B. The length of limbs of the 6-week-old *Mast4*^{+/+} and *Mast4*^{-/-} mice using µCT images.

Fig. C. Bone formation analysis of 3-week-old *Mast4*^{+/+} and *Mast4*^{-/-} mice. Bone formation was visualized by double calcein labelling at 7d and 2d prior to sacrifice, and the distance between labels were measured at 3 points per tibia (n=10 tibias for *Mast4*^{+/+} and 5 tibias *Mast4*^{-/-} mice).

Fig. D. The mRNA and protein expression of Mast4 in murine organs. (a) The qRT-PCR analysis (b) WB analysis of Mast4 expression.

5. Along the lines of point #4, quantification of the bone mass results shown in Figure 5f is needed. These graphs have no error bars or statistical significance. Also- analysis of cortical bone is needed to better understand the changes seen in trabecular bone.

As Reviewer #2 pointed out, we measured trabecular bone volume and bone mineral density (BMD) with increased number of samples. The tibias of 6-week-old male mice (n=8 for *Mast4*^{+/+} and 10 for *Mast4*^{-/-} mice) were measured. Consistently, both the trabecular bone volume and BMD in the tibias of *Mast4*^{-/-} mice were significantly lower than those of *Mast4*^{+/+} mice (see **Fig. E**). Incidentally, we realized the BMD unit was mistakenly as labeled as mg/cm³ and corrected it to g/cm³. In addition, following Reviewer #2's suggestion, we measured the cortical bone volume, BMD, and thickness from the same mice and observed their reduction in the cortical bone of *Mast4*^{-/-} mice as shown in the trabecular bones. **We removed the original graphs in Fig. 5f, added the new ones in Fig. 5e and**

added graphs analyzing cortical bone to EDF 20b.

Fig. E. The analyses of trabecular and cortical bones in the tibias of *Mast4*^{+/+} and *Mast4*^{-/-} mice at 6 weeks of age. (a) Trabecular bone volume and BMD (b) Cortical bone volume, BMD and thickness.

6. The final experiment using human cells transplanted into rabbits is quite important and interesting. The authors should move the quantification of these results into the main text figures. Each animal should be shown as an independent data point (rather than the box/plunger plots shown).

As Reviewer #2 suggested, we moved the graphs presenting the scores of total cartilage repair and thickness of reparative cartilage to Fig. 7d from EDF 29 (previously EDF 28), which showed a significant difference between naïve- and MAST4-depleted hBMSC-injected groups (see Fig. F). Then, we moved gross and microscopic images of low magnification from Fig. 7c-7g to EDF 28 (previously EDF 27). We also changed the graphs from the box/plunger plots to scatter plots to show independent data points (Fig. 7d, EDF 28 and 29; see Fig. G).

Fig. F. Histological grading of the regenerated cartilage at the full-thickness articular cartilage

defect sites in rabbit knee. The graphs in a red box were moved to the main text figures.

Fig. G. Quantitative evaluation of the percentage of tissue area stained with Masson's trichrome and type II collagen.

7. Along these lines, have the authors measured articular chondrocyte thickness in the *Mast4*^{-/-} mice? It seems that their current analyses are restricted to growth plate chondrocyte measurements.

Following Reviewer #2's suggestion, we measured tibial articular cartilage thickness in *Mast4*^{+/+} and *Mast4*^{-/-} mice at 6 weeks of age (n=6). In the safranin O-stained tissues, we could not observe a significant difference in the thickness of articular cartilage between *Mast4*^{+/+} and *Mast4*^{-/-} mice (see Fig. H). However, we found that safranin O-positive area, which staining intensity is known to be proportional to the proteoglycan content in normal cartilage, was significantly increased in the tibial articular cartilage of *Mast4*^{-/-} mice. On the other hand, we observed that *Mast4*^{-/-} mice exhibited unilaterally torsional tooth malocclusion between the maxillary and mandibular incisors (*a manuscript regarding dental malocclusion observed in Mast4*^{-/-} mice is currently under review in another journal). So, starting at 7-8 weeks of age, we had to cut their incisors every 10 days to help their food intake. As we anticipate the difference in articular cartilage thickness will be more pronounced in older mice, we are currently mating *Mast4*-flox mice with *Col2a1*-Cre mice to generate cartilage-specific *Mast4* conditional knockout mice. Using these mice that do not exhibit malocclusion (*we hope so*), we expect to be able to investigate the effect of *Mast4* deficiency on articular cartilage in aged mice. Therefore, we would like to further investigate the importance of *Mast4* in cartilage in future studies.

Fig. H. Histomorphometric analyses of tibial articular cartilage in *Mast4*^{+/+} and *Mast4*^{-/-} mice at 6 weeks of age. (a) Representative images of safranin O staining (n=6). (b) The average thickness of total tibial articular cartilage was measured at 3 points per tibia. (c) The safranin O-positive area was quantified by Image J using identical thresholding parameters between all images and detecting the total purple-red pixel area (safranin O-positive).

Minor points:

1. For EDF3, quantification of the Sox9 mRNA and protein levels are needed to support the claim that Mast4 preferentially regulates the protein.

Following Reviewer #2's suggestion, the band intensities representing Sox9 protein expression levels were quantified as the ratio of Sox9 to β -actin using densitometry with the ImageJ software (see Fig. I). As to Sox9 mRNA level, we conducted qRT-PCR anew and replaced the original blot with the graph. We presented these quantified graphs in EDF3.

Fig. I. Sox9 expression in wild-type and Mast4-depleted C3H10T1/2 cells. (a) Endogenous Sox9 protein expression. (b) The mRNA expression of Sox9 was examined by qRT-PCR.

2. For EDF9a, it would be ideal to show Sox9 ChIP in Mast4 overexpressing cells. For these ChIP studies, have the authors considered effects of Mast4 on total cellular Sox9 levels?

As Reviewer #2 recommended, we repeated Sox9 ChIP assay, including wild-type, Mast4 KO#1 and #2, Mast4-PDZ-overexpressing, and shMast4#1 and #2 C3H10T1/2 cells differentiated to chondrocytes. As expected, Sox9 binding to *Col2a1* gene was increased in CRISPR/Cas9-mediated Mast4 knockout or shRNA-mediated Mast4 knockdown cells, while it was decreased in Mast4 overexpressing cells (see Fig. J). We replaced the original graph in EDF9a with the new Sox9 ChIP assay results shown below. Regarding the effects of Mast4 on total cellular Sox9 levels, although we presented the results of Sox9 binding to *Col2a1* gene before we showed Mast4 regulation of Sox9 protein expression, we do consider that the different degrees of Sox9 binding in Mast4-depleted or overexpressing cells may be attributed to the total Sox9 protein expression levels regulated by Mast4 status.

Fig. J. Sox9 ChIP on the *Col2a1* gene using the indicated differentiating C3H10T1/2 cells.

3. For EDF9c and EDF9d, quantification of immunoblotting results is needed, especially since MG132 seems to increase Sox9 protein levels even in cells without Mast4 over-expression

According to Reviewer #2's suggestion, the band intensities representing Myc-Sox9 protein

expression levels were quantified as the ratio of Sox9 to β -actin using densitometry with the ImageJ software (see Fig. K). As to EDF9c, we replaced the original western blot result with the new one including Mast4 siRNA to support that effects of truncated Mast4 overexpression are dose-dependent and not due to alterations of protein activity. And as Reviewer #2 pointed out, we acknowledge the increase of Sox9 protein level by MG-132 in the absence of Mast4. In fact, in repeated experiments, we found that the basal expression level of transiently overexpressed Myc-Sox9 and HA-Mast4-PDZ were often increased by MG-132 treatment alone. Based on these observations, we consider that both Sox9 and Mast4 may be susceptible to proteasomal degradation by various proteins. Indeed, it was reported that E3 ligase Fbw7 induces Sox9 degradation (Hong et al. Nucleic Acids Res., 2016). Also, we consider that Mast4 stability may be regulated by other unknown factors other than Smurf1/GSK-3 β .

Fig. K. Mast4 regulation of Sox9 stability. (a) Full-length (HA-Mast4-FULL) and truncated Mast4 (HA-Mast4-PDZ) together with Mast4 siRNA (Santa Cruz Biotechnology; sc-106201) that targets transfected Mast4 plasmid (human) were co-transfected with Myc-Sox9 into C3H10T1/2 cells. (b) Myc-Sox9 and HA-Mast4-PDZ were transfected to C3H10T1/2 cells, followed by MG-132 treatment.

4. For Figure 2E, the authors cannot claim direct protein/protein interaction based on cell-based co-immunoprecipitation results. To do this, recombinant proteins are needed

As Reviewer #2 pointed out, we removed the word 'directly' and **edited the sentence** "Mast4 directly interacted with Sox9 protein" to "Mast4 bound to Sox9" (p.6).

5. Figure 2G does not include MG132 treatment even though the text indicates that it does

We apologize for the confusion. The statement of MG132 treatment refers to EDF9e (previously EDF9c). To remove confusion, **we rearranged Fig. 2g to Fig. 2f** and coupled it with EDF9c showing Mast4-mediated Sox9 degradation in a dose-dependent manner, presenting it before reduced Sox9 serine phosphorylation in Mast4-depleted cells (previously Fig. 2f, currently rearranged to Fig. 2g). Later, **we mention separately** that MG-132 treatment restored Mast4-induced Sox9 degradation **in EDF9e** (previously EDF9d). **Changes are shown in p.6.** We thank Reviewer #2 pointing out the confusion and allowing us to make it more straightforward.

6. For EDF10A, the changes in Sox9 S181 and S199 phosphorylation are quite modest. Why do the authors think that these changes are 'real'? Were replicates performed?

To identify Mast4-mediated Sox9 phosphorylation sites, two separate MASS SPEC analyses were conducted. As Reviewer #2 mentioned, we observed a very modest increase of Sox9 S181 and S199 phosphorylation level in the presence of Mast4 from two trials. Moreover, unlike Sox9 S494A mutation, neither Sox9 S181A nor S199A mutation showed evident increase of protein stability in the presence of Mast4. We consider that the readers may raise a question as Reviewer #2 did and since we have not expanded the study on Mast4 regulation of Sox9 S181 or S199, **we have decided to remove EDF10a by presenting Fig. 2h alone which introduces Sox9 S494 phosphorylation.**

7. For Figure 2I- it is not totally clear why the S494D phospho-mimetic form of Sox9 still responds to Mast4 overexpression. The authors should provide a potential explanation for this finding.

As Reviewer #2 pointed out, we also acknowledge that Sox9 S494D is still regulated by Mast4-overexpression, although the extent was less than that shown in the wild-type Sox9. Apart from Mast4 post-translational regulation of Sox9, we demonstrated not only the interaction between Mast4 and Smad3, but also increased transcriptional activity of Smad3, a key mediator of TGF- β signaling, due to Mast4 depletion (EDF 11). Since it has been reported that TGF- β itself stabilizes Sox9 through post-translational modification^{1,2}, the mechanism by which Mast4 directly regulates Sox9 through serine 494 phosphorylation and the mechanism of indirectly regulating Sox9 through TGF- β signaling, regardless of S494 phosphorylation, may be still valid, resulting in modest response of Sox9 S494D response to Mast4.

¹ Coricor G, Serra R. TGF- β regulates phosphorylation and stabilization of Sox9 protein in chondrocytes through p38 and Smad dependent mechanisms. *Sci Rep.* 2016 Dec 8;6:38616. doi: 10.1038/srep38616.

² Chavez RD, Coricor G, Perez J, Seo HS, Serra R. SOX9 protein is stabilized by TGF- β and regulates PAPSS2 mRNA expression in chondrocytes. *Osteoarthritis Cartilage.* 2017 Feb;25(2):332-340. doi: 10.1016/j.joca.2016.10.007. Epub 2016 Oct 13.

8. For EDF21B, the authors should provide quantitative measurement of their back-scatter electron imaging results. Alternatively, the authors could simply perform cortical bone microCT results to measure cortical thickness and cortical bone tissue mineral density.

According to Reviewer #2's suggestion, we measured cortical bone volume, bone mineral density and thickness from μ CT images of the tibias of 6-week-old male mice (n=8 for *Mast4*^{+/+} and 10 for *Mast4*^{-/-} mice), as previously demonstrated in Reviewer #2's 5th major concern. Similar to BEI images of EPMA data in EDF20c (previous EDF21b), the quantitative values of cortical bone were decreased in the tibias of *Mast4*^{-/-} mice. **We added the graphs analyzing cortical bone to EDF 20b.**

Reviewer #3:

Major concerns:

1. The introduction needs some introduction to Mast4 and its prior description in the literature. It is known to have a role in a few osteoblast relevant signaling pathways (e.g. FGF2, per Lee et al., Cell Death Diff, 2021).

First, according to Reviewer #3's suggestion along with Minor Concern #1, **we revised the introduction** by mentioning the importance of studying protein kinases in signal transduction, which was the background/rationale of our study on Mast4 (p.3). Furthermore, **we included a brief introduction of the prior description regarding the role of Mast4** associated with the FGF2/ERM pathway in helping self-renewal of spermatogonial stem cells **in the discussion section (p.14)** after examining FGF2 regulation of Mast4 in C3H10T1/2 cells as described below in Major Concern #2. Unlike previous result showing FGF-2-induced Mast4 transcription in Tm4 human Sertoli cells, mRNA expression of Mast4 was not regulated by FGF-2 in C3H10T1/2 cells. This indicates FGF-2 regulation of Mast4 transcription may be cell-type-specific. Also, while previous results claim the role of Mast4 in self-renewal of spermatogonial stem cells, we focus on the role of Mast4 in the differentiation of MSCs. In addition, even though FGF-2 has been reported to promote osteogenesis and bone formation, we could not observe FGF-2 regulation of Mast4 in C3H10T1/2 cells, indicating that Mast4 particularly mediates Wnt-induced osteogenesis of MSCs. Based on these considerations, we decided not to mention the relationship between FGF-2 and Mast4 in the introduction but solely in the discussion section. We hope Reviewer #3 understand our intention to avoid confusion and to highlight the role of Mast4 on MSC differentiation.

2. Related to the prior point, it is important to look at and address previous roles that may potentially be affecting the currently described role.

According to Reviewer #3's advice, we first examined whether mRNA or protein expression of Mast4 was regulated by FGF-2 in C3H10T1/2 cells (see Fig. A). Unlike in Tm4 human Sertoli cells, the mRNA expression of neither Mast4 nor ERM and its target genes was regulated by FGF-2 in C3H10T1/2 cells. The protein expression of Mast4 was not regulated by FGF-2 as well in C3H10T1/2 cells. In addition, Mast4 regulation of ERM phosphorylation was not shown in C3H10T1/2 cells, indicating that FGF-2 regulation of Mast4 transcription and post-translational regulation of ERM by Mast4 might be limited to Sertoli cells.

Fig. A. FGF-2 regulation of Mast4 transcription and Mast4 regulation of ERM in C3H10T1/2 cells. (a) RT-PCR examination of *Mast4*, *Erm* and *Erm* target genes by FGF-2 in C3H10T1/2 cells. (b) Mast4 protein expression by FGF-2 in C3H10T1/2 cells. (c) Western blot analysis of ERM serine phosphorylation by Mast4 in C3H10T1/2 cells.

3. Figure 2 needs shRNA knockdown of Mast4 to confirm that the effects of genomic knockout aren't unique to the particular knockout used.

4. This shRNA knockdown needs to be included for all key experimental findings where other knockdown/knockout methods are used – especially the experiments showing that Mast4 KO results in increased cartilage formation (Figure 2) and decreased osteoblastogenesis (Figure 4).

As Reviewer #3 suggested, we established 2 different shRNA-mediated Mast4 knockdown C3H10T1/2 cell lines and repeated key experiments again by including these cell lines (see **Fig. B**). Similar to CRISPR/Cas9-mediated Mast4 knockout C3H10T1/2 cells, these cell lines showed increased mRNA expression of chondrocyte marker genes (added to **EDF2d**), chondrogenic differentiation (**EDF7c**), Sox9 binding to Col2a1 gene (**EDF9a**) and Sox9 protein levels (**EDF9d**), but decreased Sox9 serine phosphorylation (**EDF9d**) and osteogenic differentiation (**EDF15b**). We added these results to **Extended Data Figures** as indicated above and mentioned about establishment of Mast4 knockdown C3H10T1/2 cell lines in the manuscript (p.5).

Fig. B. Examination of shRNA-mediated Mast4 knockdown C3H10T1/2 cell lines. (a) The mRNA expression of chondrocyte marker genes in undifferentiated control and shMast4 #1, #2 C3H10T1/2 cells. (b) Sox9 ChIP on *Col2a1* gene in differentiating C3H10T1/2 cell lines. (c) Mast4-induced Sox9 serine phosphorylation in differentiating C3H10T1/2 cell lines. (d) Chondrogenic and (e) osteogenic differentiation of the indicated cell lines were examined by alcian blue and Alizarin Red S staining, respectively.

5. Figure 5 needs microCT confirmation of changes in cartilage and hexabrix analysis is particularly important for demonstrating changes in cartilage with the knockout of Mast4.

As Reviewer #3 suggested, it would be more compelling if we could demonstrate changes in articular cartilage with the knockout of Mast4 *in vivo*. However, unfortunately, hexabrix was discontinued two years ago and is no longer in production by Mallinckrodt Inc. We inquired about the remaining products, but failed to find any to conduct an analysis. We hope that Reviewer #3 would

understand the reason for absence of data showing changes in articular cartilage. Instead, we measured tibial articular cartilage thickness in *Mast4^{+/+}* and *Mast4^{-/-}* mice at 6 weeks of age (n=6). In the safranin O-stained tissues, we could not observe a significant difference in the thickness of articular cartilage between *Mast4^{+/+}* and *Mast4^{-/-}* mice. However, we found that safranin O-positive area, which staining intensity is known to be proportional to the proteoglycan content in normal cartilage, was significantly increased in the tibial articular cartilage of *Mast4^{-/-}* mice (see Fig. C). On the other hand, we observed that *Mast4^{-/-}* mice exhibited unilaterally torsional tooth malocclusion between the maxillary and mandibular incisors (*a manuscript regarding dental malocclusion observed in Mast4^{-/-} mice is currently under review in another journal*). So, starting at 7-8 weeks of age, we had to cut their incisors every 10 days to help their food intake. As we anticipate the difference in articular cartilage thickness will be more pronounced in older mice, we are currently mating Mast4-flox mice with Col2a1-Cre mice to generate cartilage-specific Mast4 conditional knockout mice. Using these mice that do not exhibit malocclusion (*we hope so*), we expect to be able to investigate the effect of Mast4 deficiency on articular cartilage in aged mice. Therefore, we would like to further investigate the importance of Mast4 in cartilage in future studies.

Fig. C. Histomorphometric analyses of tibial articular cartilage in *Mast4^{+/+}* and *Mast4^{-/-}* mice at 6 weeks of age. (a) Representative images of safranin O staining (n=6). (b) The average thickness of total tibial articular cartilage was measured at 3 points per tibia. (c) The safranin O-positive area was quantified by Image J using identical thresholding parameters between all images and detecting the total purple-red pixel area (safranin O-positive).

6. Figure 5 also needs confirmation of changes in bone formation using time separated dual staining with Calcein (and/or Alizarin red) injection to show changes in bone growth between WT and Mast4 knockout mice in vivo, e.g. Rauch et al. (PLoS One, 2010) or Su et al. (JCI, 2012).

As Reviewer #3 recommended, we examined bone formation of *Mast4^{+/+}* and *Mast4^{-/-}* mice at 3 weeks of age through time separated staining of Calcein as conducted in the papers suggested. Calcein was injected to the mice with an interval of 5 days. The mice were sacrificed 2 days after 2nd injection, and the gap width of calcein stained on the outer border of tibial cortical bone was measured at 3 points per tibia. Ten tibias for *Mast4^{+/+}* mice and 5 tibias for *Mast4^{-/-}* mice were used. We observed decreased bone formation in *Mast4^{-/-}* mice (see Fig. D). **The results were added to Fig. 5f and mentioned in the manuscript (p.10).** We would like to express our appreciation for Reviewer #3's suggestion that led to the improvement of our data.

Fig. D. Bone formation in the tibias of *Mast4*^{+/+} and *Mast4*^{-/-} mice visualized by double calcein labelling.

7. Figure 6 needs qPCR (e.g. SYBRgreen) with quantification of multiple replicates rather than RT-PCR for this critical analysis and confirmation.

As Reviewer #3 suggested, we performed qRT-PCR using cartilage and bone tissues isolated anew from *Mast4*^{+/+} and *Mast4*^{-/-} mice at PN 1 day and **replaced the original RT-PCR blots (previously Fig. 6d, 6e) with new qRT-PCR graphs (EDF26b, 26d) (see Fig. E).** Since we also presented certain protein expressions of these cartilage and bone tissues as suggested below in Comment No.9, we rearranged the qRT-PCR data from the main figure to EDF26.

Fig. E. The qRT-PCR data of (a) Sox9 target and (b) osteogenesis-associated genes in the mouse cartilages and bones, respectively, of *Mast4*^{+/+} and *Mast4*^{-/-} mice at PN 1 day.

8. The truncated overexpression of *Mast4* should have more justification (beyond “Due to high molecular weight (>285 kDa) and relatively low expression of full-length *Mast4*, we alternatively used a truncated *Mast4* construct (*Mast4*-PDZ)” and extended data figure 4) and/or shRNA-knockdown should be used to know that effects are dependent on *Mast4* dose.

We appreciate Reviewer #3’s insightful comment concerning the provision of controls to support our conclusions that the effects of truncated *Mast4* overexpression are due to dose and not to alterations of protein activity. We reinforced the original result (previously EDF9c) by transfecting HA-*Mast4*-PDZ (truncated) or HA-*Mast4*-FULL (full-length) in a dose-dependent manner together with *Mast4* siRNA (Santa Cruz Biotechnology; sc-106201) that targets transfected *Mast4* plasmid (human). Due to inefficient detection of full-length *Mast4* with HA antibody, we used the *Mast4* antibody that recognizes the C-terminus. In addition, we examined the transfection efficiency of *Mast4*-PDZ or

Mast4-FULL plasmids in a separate experiment by RT-PCR for the mRNA expression of human *MAST4*, as shown below (Fig.6). We observed that both full-length and truncated Mast4 induced Sox9 degradation in a dose-dependent manner, while inhibition of Mast4 by siRNA suppressed Sox9 degradation in C3H10T1/2 cells (see Fig. F and G). This result indicates that the degree of Mast4 dose may be sufficient for Mast4 regulation of Sox9. **We replaced the original western blot result (previously EDF9c) with the new result and mentioned it in the manuscript (p.6)**

Fig. F. Sox9 is regulated by both truncated and full-length Mast4 in C3H10T1/2 cells.

Fig. G. The RT-PCR examination of transfection efficiency.

9. Western blot should be used to detect the protein level changes and signaling pathways changes in mutant mice compared with Wild-Type mice, including genes associated with cartilage development in the cartilage of *Mast4*^{-/-} mice and genes associated with skeletal system development and Wnt signaling pathway in the bone of *Mast4*^{-/-} mice compared that in *Mast4*^{+/+}.

As Reviewer #3 suggested, proteins were extracted from the cartilage and bone tissues of 1-week-old *Mast4*^{+/+} and *Mast4*^{-/-} mice, followed by western blotting (see Fig. H). We found that two key markers of chondrocytes, Sox9 and Col2a1, were increased in the cartilages of *Mast4*^{-/-} mice. In addition, we observed that the total level of not only Runx2 and Mmp13, which is the target of Runx2, but also β -catenin, a key Wnt signal mediator, was considerably reduced in the bones of *Mast4*^{-/-} mice. **We added this result to Fig. 6d, 6e, and it is mentioned in the manuscript (p.11)**

Fig. H. The expression of key proteins associated with (a) cartilage development and (b) bone development/Wnt signaling in the mouse cartilages and bones, respectively, of *Mast4*^{+/+} and *Mast4*^{-/-} mice at 1 week of age.

Minor concerns:

1. *The abstract seems to have obviously been written too quickly and should include at least a sentence providing background by answering the question, “Why are we looking at Mast4?”*

As Reviewer #3 suggested, **we revised the abstract** (1) by stating that protein kinases, including Mast4, play a crucial role in signal transduction, and (2) by rephrasing sentences explaining the roles of Mast4 as a mediator of TGF-β-induced chondrogenesis and Wnt-induced osteogenesis of MSCs.

2. *Figure legends shouldn’t include results, simply descriptions of what is being done. Descriptions of results should be limited to the results and figure legend titles.*

We edited the figure legends by excluding any description of the results.

3. *Primer sequences need to be provided for each RT-PCR and/or any qPCR performed.*

A list of primers used in this study is provided in Supplementary Table 1.

4. *Antibody catalog numbers and/or clone IDs need to be provided for the antibodies used in the western blots performed.*

A list of antibodies used in this study is provided in Supplementary Table 1.

REVIEWERS' COMMENTS

Reviewer #1 (Remarks to the Author):

No further comments

Reviewer #2 (Remarks to the Author):

The authors have addressed all previous comments in a satisfactory manner. The revised manuscript is much improved and will be of interest in the bone and cartilage field.

Reviewer #3 (Remarks to the Author):

The authors have adequately addressed reviewer questions and concerns. The manuscript now is ready to be published in Nature Communications